# Nucleotides Entrapped in Liposome Nanovesicles as Tools for Therapeutic and Diagnostic Use in Biomedical Applications

**DOI:** 10.3390/pharmaceutics15030873

**Published:** 2023-03-08

**Authors:** Camila Magalhães Cardador, Luis Alexandre Muehlmann, Cíntia Marques Coelho, Luciano Paulino Silva, Aisel Valle Garay, Alexandra Maria dos Santos Carvalho, Izabela Marques Dourado Bastos, João Paulo Figueiró Longo

**Affiliations:** 1Department of Genetics and Morphology, Institute of Biological Sciences, University of Brasília (UnB), Brasilia 70910-900, DF, Brazil; 2Faculty of Ceilandia, University of Brasilia, Brasilia 70910-900, DF, Brazil; 3Laboratory of Synthetic Biology, Department of Genetics and Morphology, Institute of Biological Science, University of Brasília (UnB), Brasilia 70910-900, DF, Brazil; 4Laboratório de Nanobiotecnologia (LNANO), Embrapa Recursos Genéticos e Biotecnologia, Brasilia 70770-917, DF, Brazil; 5Molecular Biophysics Laboratory, Department of Cell Biology, Institute of Biological Science, University of Brasília (UnB), Brasília 70910-900, DF, Brazil; 6Pathogen-Host Interface Laboratory, Department of Cell Biology, University of Brasilia (UnB), Brasilia 70910-900, DF, Brazil

**Keywords:** liposome, nucleotide, protection, therapeutic, diagnostic

## Abstract

The use of nucleotides for biomedical applications is an old desire in the scientific community. As we will present here, there are references published over the past 40 years with this intended use. The main problem is that, as unstable molecules, nucleotides require some additional protection to extend their shelf life in the biological environment. Among the different nucleotide carriers, the nano-sized liposomes proved to be an effective strategic tool to overcome all these drawbacks related to the nucleotide high instability. Moreover, due to their low immunogenicity and easy preparation, the liposomes were selected as the main strategy for delivery of the mRNA developed for COVID-19 immunization. For sure this is the most important and relevant example of nucleotide application for human biomedical conditions. In addition, the use of mRNA vaccines for COVID-19 has increased interest in the application of this type of technology to other health conditions. For this review article, we will present some of these examples, especially focused on the use of liposomes to protect and deliver nucleotides for cancer therapy, immunostimulatory activities, enzymatic diagnostic applications, some examples for veterinarian use, and the treatment of neglected tropical disease.

## 1. General Background of Liposomes

### 1.1. Definitions

Liposomes are nano- or microvesicles formed by single or multiple bilayers of lipids—usually a mixture of phospholipids and cholesterol—that are self-assembled in aqueous media by hydrophobic interactions [1]. Different types of liposomes, which differ in morphology and size, are described in the literature (Figure 1) [2]. Unilamellar vesicles (ULVs) are liposomes with a single lipid bilayer enclosing an aqueous core. ULVs can be subdivided into small unilamellar vesicles (SUVs), ranging from 20 to 100 nm in diameter, large unilamellar vesicles (LUVs), with diameters between 100 and 1000 nm, or giant unilamellar vesicles (GUVs), with diameters above 1 µm [3]. Oligolamellar vesicles (OLVs) are formed by two to five, while multilamellar vesicles (MLVs) present more than five concentric lipid bilayers [3]. An additional class is formed by multivesicular liposomes (MVLs), which are multiple, nonconcentric lipid bilayer vesicles all enclosed by a single lipid bilayer [2].

Due to phospholipid bilayer biochemistry, two different nanoscopic compartments are formed in the liposome vesicles. One is the aqueous central compartment, where hydrophilic molecules are placed and carried inside this nanoscopic space. The other is the hydrophobic space in the bilayer interface, where the insoluble molecules are entrapped and protected among the phospholipid fatty acids. Thus, two different compartments are created with different biochemical characteristics and allow the encapsulation of a variety of molecules of interest (Figure 1). Moreover, liposomes are easily prepared and there are some examples of industrial scaled-up processes that allow their incorporation into innovative nanotech products. That is why this lipid carrier has been the most studied and investigated nanostructure for biomedical applications. For nucleotide liposomes, the DNA or RNA molecule can be entrapped in the inner aqueous space or be absorbed on the vesicle surface due to the cationic phospholipids charges.

In terms of applications, liposomes have been used in different biomedical areas, including pharmacy, oncology, immunology, cosmetology, and diagnosis. Historically, liposomes have been used to modify the behavior of carried molecules, as well as to protect these molecules carried from the environment. Regarding the use of nucleotides, as carried molecules, liposomes are used [1] to maintain the integrity of nucleotide sequences, to increase their half-life in biological media, and [4] to facilitate their transport across biological membranes, thus optimizing the pharmacokinetic profile of these biomolecules. For this third application, it is important to highlight that as hydrophilic and polar molecules, the nucleotides usually have a low transport rate through biological membranes [5].

Moreover, in comparison to other nucleotide transporters, such as viral vectors, lipid nanocarriers have some additional advantages (Figure 2). In terms of immunogenicity, liposomes, or alternative lipid carriers, may have reduced immunogenicity, in comparison to these conventional viral carriers [5]. The development of these nanocarriers was also a response to the replacement of these other types of nucleotide carriers, which normally go through biotechnological manufacturing routes. For instance, the use of viral vectors may also contain highly immunogenic moieties that can induce adverse immunogenic reactions. This concern is also extended to lipid nanocarriers. At the moment, the liposome immunogenicity seems to be related to the presence of molecules such as polyethylene glycol (PEG), which could be replaced in a near future by low immunogenic molecules. Additionally, it is possible to replace potential immunogenic lipid components or even remove them from the particle formulation.

Moreover, in terms of industrial manufacturing, liposomes can be easily prepared in conventional industrial protocols. On the other hand, viral vectors are normally prepared in biotechnology industrial routes, which can impact the scale-up process. Moreover, biotechnology pathways have a larger risk of nonintentional contamination. Thus, we understand that lipid nanocarriers, such as liposomes, are a suitable nanotech alternative for nucleotide delivery to target cells during theranostic applications.

### 1.2. Historical Aspects

In terms of historical landmarks, liposomes were originally developed in the mid-1960s, when some researchers observed the formation of membrane-like structures when they hydrated dried phospholipid films. Due to these observations, and their phospholipid constitution, these artificial vesicles were initially used as biomimetic membranes [4]. Thus, their first use was as a cell membrane model for biophysics and membrane transport studies. This initial liposome use was not the final one for these vesicles but was fundamental to understanding how this artificial phospholipid membrane works. Furthermore, these initial steps were fundamental to establishing the protocols for the preparation and stabilization of these lipidic nanostructures.

Sometime after that, the same research group that proposed liposomes as a biomimetic membrane model identified that they could be used to entrap enzymes and proteins and modify their fate when administered to experimental animals [6]. These first observations triggered the development of a new investigation area in pharmacology. It was an important technical step forward for these targeted delivery approaches. Thus, since then, thousands of articles were published on this topic. Most of them had a specific focus to modulate the delivery of encapsulated molecules inside the body.

Therefore, it is not by chance that the first nanotechnological products applied to human beings were those technologies involving the use of liposomes. First, in the 1980s, the cosmetic industry used liposome vesicles to entrap and protect cosmetic active ingredients, such as ascorbic acid. In this case, the liposome was used to protect and extend the shelf life of ascorbic acid, a well-known unstable molecule. This first application was a topical one and opened the way for the development of more complex parenteral liposome administrations [7].

One decade later, the regulatory agencies approved the use of the first pharmacological liposome, Doxil™, for some oncological conditions, such as ovarian cancer and Kaposi sarcoma. For Doxil™, the idea was a little different. At that time, researchers observed that Doxil™, a liposome containing the chemotherapeutical drug doxorubicin, promoted a different drug biodistribution profile. Moreover, the toxicity related to doxorubicin, especially cardiac toxicity, was significantly reduced when the drug was associated with liposomes [8,9,10,11].

After Doxil™, hundreds of projects involving nano-sized liposomes were developed [12]. The main idea was to reformulate traditional drugs, reduce undesirable side effects, and eventually improve the delivery of these conventional drugs. Thus, almost all new liposomes released onto the market were a result of research and development of a new formulation of a well-known drug. This is quite an interesting approach because well-known drugs do not need new approvals, since they already are in use and on the market [13,14]. Regulatory approval will be carried out only for the drug carrier, which is a less risky process in terms of regulatory barriers.

After Doxil™ approval, governmental funding agencies, as well as private companies, invested a lot of effort in developing new nanomedicines for different medical conditions. Most of the efforts aimed to develop nanotech drugs for oncological conditions [10,11], but other diseases, such as infectious disease [15] and autoimmune conditions, were also the target of research into nanomedicines. In terms of scientific publications, thousands of articles were published, hundreds of clinical trials were conducted, and at least 15 nanoparticle-based drugs were approved worldwide. This is a very important outcome, since the development of a new drug is extremely expensive, reaching billions of dollars for each new single drug approved and released onto the market [16].

More recently, in 2018, the first lipid nanoparticle containing nucleotides was approved by the Food and Drug Administration (FDA) for amyloidosis treatment [13,17]. Despite not being a true liposome, the development of lipid nanocarriers has increased the understanding of this type of nanostructure for the protection and delivery of nucleotides at target sites. This product, named Patisiran/Onpattro™, is composed of a siRNA entrapped in a cationic lipid nanoparticle that was originally developed to extend the shelf life of the RNA sequence. Moreover, the lipid nanoparticles were also applied to deliver the sequence to the target tissues. This nanotech product is used to treat hereditary transthyretin amyloidosis, a genetic autoimmune disease caused by mutations in the transthyretin gene. As the main source of this altered transthyretin is the liver, which releases the altered proteins into the bloodstream, the use of a cationic liposome to deliver the siRNA is an interesting strategy, since these nanoparticles are passively absorbed by the liver tissue. In terms of pharmacodynamics, the siRNA released in the liver should target the transthyretin mRNA, thus reducing the translation of this mutated protein [18,19,20].

The most recent and important point in this liposome–nucleotide journey was the development of mRNA vaccines for COVID-19 [7,18]. These liposome–mRNA vaccines, which were approved by different regulatory agencies worldwide, have been by far the nanomedical product most widely used by humans. The development of these vaccines was the result of the combination of two important technological approaches. First, the idea to use nucleotides as therapeutical biomolecules was proposed during the 1980s, but not implemented due to the typical instability of these molecules in biological media. The second was to use lipid nanocarriers to protect and deliver these active ingredients to target cells and tissues [21]. For instance, nucleotide molecules are not well delivered to cells due to their high hydrophilicity. The encapsulation of these molecules in liposomes can improve this delivery, allowing them to reach the inner compartments of live cells. In Figure 3, we present a representative scheme with this process. While free molecules are not well absorbed, nucleotides entrapped in liposomes can pass through the membrane more easily.

During the pandemic, a part of society was surprised and impressed with this innovative approach. A lot of people questioned the speed of development of this technology. However, the innovation process had begun in the 1960s, when the first liposomes were developed as biomimetic biological membranes. Moreover, several validation stages had been overcome for other therapeutical purposes over recent decades. In addition, during the pandemic, researchers and the industry had a lot of information regarding the viability of this approach to produce a feasible vaccine against SARS-CoV-2 infection. It is interesting that for the general public, it may seem that technologies are developed in a kind of “Eureka” moment, but in reality, they arise from a long process built up by numerous researchers and innovators over time. For this specific case, over decades.

The last part of this historical review involves the increase in published articles using nucleotides and liposomes. The successful combination of liposomes that protect and deliver the nucleotide sequences has been extensively proposed for very different applications. These technologies have been suggested for both therapeutic and diagnostic use, and different medical conditions, including immunization for cancer and infections, oncology therapy, and diagnosis of different types of disease. During the next sections, we will review the general aspects of this innovative approach, as well as revise how they have been used for markedly different medical conditions. For sure, the next decades will produce many successful examples demonstrating the use of nucleotides entrapped in liposomes for biomedical applications.

### 1.3. Cationic Liposomes for Nucleotide Encapsulation

As described before, liposomes are formed by a mixture of phospholipids, but it is common to use cholesterol molecules to improve the vesicle stability over time. Other constituents were also evaluated in liposome formulation, and probably, the most important one was the polyethylene glycol polymer (PEG). These PEG molecules were added to the liposome surface, usually attached to the phospholipid molecules, to increase their hydrophilicity, thus reducing the potential vesicle aggregation. Moreover, PEGylated liposomes have other interesting properties, such as long circulation time after intravenous administration. The reason for this property is the reduction in plasma protein deposition over the liposome surface, an event that impairs vesicle recognition by the immune system, thus increasing the liposome circulation time [1,22,23].

In terms of biochemistry, liposome phospholipids are defined as natural or synthetic lipids formed by two fatty acid hydrophobic tails, an intermediary alcohol group, such as glycerol, and a polar phosphate headgroup. In this phospholipid organization, it is possible to observe an amphiphilic pattern, which is fundamental for liposome vesicle preparation. Actually, due to this amphiphilic characteristic, these phospholipids are first organized in thin lipid films, and they are self-assembled in heterogeneous liposome vesicles, with different sizes and layers, when mixed with aqueous media. In this way, the vesicle is formed with low-energy protocols. The energy used is obtained by hydrophobic repulsion forces that promote the phospholipid orientation with the phosphate polar head oriented to the external aqueous media, while the apolar fatty acids are oriented to the inner part of the vesicle wall bilayer. To prepare nanoscopic vesicles, other steps, such as sonication and vesicle extrusion, are applied to obtain regular nano-sized structures.

In terms of composition, we can cite the glycerophospholipids, phosphatidylcholine and phosphatidylcholine, and the sphingolipid sphingomyelin [2] as examples of widely used phospholipids. These lipids present two characteristics that are particularly important for the formation of vesicles: (i) amphiphilicity, i.e., their structure comprises a hydrophilic head bound to hydrophobic tails and (ii) cylindrical shape. Their hydrophobic moieties are held together inside the bilayer by hydrophobic interactions, while the hydrophilic heads interact with the aqueous media on both sides of the bilayer. Their cylindrical shape, occasioned by the similar diameters of the hydrophilic and hydrophobic moieties, renders the lipid assembly lamellar [24]. Conical lipids, such as cardiolipin and phosphatidylethanolamine, can also be added to the liposome formulation to induce curvature in the lipid bilayer [1].

The length and the saturation of the hydrocarbon chains of the acyl groups of phospholipids are important parameters in liposome formulation. They affect the interaction between vicinal lipids, thus affecting the phase transition temperature (T_C_) of the membrane [2]. The T_C_ can be defined as the temperature at which the membrane organization changes from the gel to the liquid crystalline phases. In the gel phase, adjacent hydrocarbon chains are closely associated and the membrane is more rigid, whereas in the liquid crystalline phase, they are more loosely associated and fluid [3]. Thus, a membrane will present significantly different fluidities and permeabilities, depending on its T_C_ and the temperature to which it is exposed. Generally, lipids with more saturated and/or longer acyl groups have higher T_C_. Moreover, cholesterol can be used to increase the rigidity of liposome membranes, improving their stability and affecting the release of their cargo [3,24].

The chemistry of the surface of the lipid bilayer is another crucial—and tunable—parameter in liposome formulation. The net overall charge of the hydrophilic head of phospholipids must be considered, as it can render the surface of liposomes charged or neutral at physiological pH, strongly affecting its interaction with the target surface [24]. Several lipids are available which can be used to tune the superficial Coulombic net charge of liposomes: (i) zwitterionic lipids, with a neutral net overall charge, such as phosphatidylcholine and phosphatidylethanolamine; (ii) anionic lipids, such as phosphatidylinositol and phosphatidylserine; and (iii) cationic lipids, such as dioleoyl-3-trimethylammonium propane (DOTAP). Moreover, different polymers can be used to add special attributes to the surface of liposomes. For example, as cited before, different polyethylene glycol (PEG) derivatives exist, which can be used, for instance, to increase colloidal stability, prolong the circulation time [25], and to serve as adaptors to targeting molecules, such as folic acid [22] and antibodies [26].

In most protocols for liposome preparation, the molecule encapsulation process will be executed during the vesicle’s formation. For hydrophilic compounds, the molecules are dissolved in the aqueous media, while hydrophobic molecules are dissolved in the phospholipid film from previous vesicle formation. In this situation, the encapsulation efficiency, defined as the number of molecules entrapped in the liposomes, will vary according to several aspects related to both the phospholipid composition and the nucleotide structure. In general, as hydrophilic molecules, the nucleotide will be encapsulated in the aqueous core compartment [1].

One strategy to optimize this DNA/RNA encapsulation in the liposome vesicles is to include cationic phospholipids in the liposome composition. The presence of these cationic phospholipids may attract the negative nucleotide sequences. Due to their hydrophilicity, these electrostatically bonded nucleotides will be placed inside the aqueous core or eventually absorbed over the liposome surface. That was the most frequent strategy to produce the liposome containing DNA/RNA molecules, and in addition to the high encapsulation efficiency, related to the electrostatic interaction between nucleotides and cationic liposomes, these positive vesicles may behave differentially in biological systems [27,28].

For example, these cationic liposomes interact with cell membrane surfaces, which have a residual negative charge. These negative charges are related to the presence of the phospholipid phosphate head. In this condition, the cationic liposome interacts more with cell surfaces, thus delivering the liposome-entrapped molecules faster, in comparison to the negative or neutral liposomes. This benefit can be therapeutically used to optimize nucleotide cargo, but also can reduce the bioavailability of these liposomes when administered intravenously. The mechanism is similar, and the cationic charges are quickly recognized by the plasma proteins and the immune system, which will clear the vesicles from the bloodstream [18,29].

### 1.4. Review Objectives

As will be presented in this article review, the use of nucleotides and liposomes is vast and the biomedical applications are diverse. Currently, it is not easy to preview which biomedical applications will have the best benefits from using these technologies. Thus, we understand that, at present, we should not focus on a single application or nucleotide type, but rather present the different technological approaches available in the literature. For instance, we revised the main results regarding the use of nucleotide and liposome nanocarriers in different therapeutical and diagnostic applications, including oncology and immunological conditions, neglected diseases therapies, innovative diagnostics approaches, and veterinarian applications. These application examples are summarized in the following topics (Figure 4).

## 2. Theranostic Nucleotide

As mentioned previously, the use of nucleotides for theranostic applications is not new, and several initiatives for that are presented in the literature. In this review article, we will present some of these examples, especially focused on the use of liposomes to protect and deliver nucleotides for cancer therapy, immunostimulatory activities, enzymatic diagnostic applications, some examples for veterinarian use, and the treatment of neglected tropical disease. The next sections will be dedicated to each of these examples in detail.

### 2.1. Liposomes for the Delivery of Nucleic Acids in Cancer Therapy

As described earlier, the first nanotech-based pharmaceutical product was a liposome containing doxorubicin, named Doxil™. For oncology, the use of liposomes is very common, and several examples use these lipid carriers to deliver different types of molecules, mainly chemotherapeutic drugs, such as doxorubicin, to target malignant tissues. Thus, the Doxil™ example is a hallmark in the history of liposome development.

The nucleic acid delivery is not different, and numerous papers have been published recently, aiming at this objective. Usually, the idea has been to target not only malignant cells, but also tumor-associated cells, or cells of the immune system involved in the elimination of cancer cells. These alternative targets are justified by the fact that nucleic acid can modify the behavior of these associated cells that impact the progression and tumor immune surveillance.

One target has been the expression of key effectors in signaling pathways involved in cancer progression. In 2008, for instance, Aleku et al. [30] reported preclinical data obtained with a formulation named Atu027, which consisted of liposomes loaded with siRNA directed against protein kinase N3 (PKN3) for the treatment of solid tumors. PKN3 is a downstream enzyme in the phosphoinositide-3-kinase (PI3K) signaling pathway, which is important in the process of angiogenesis. PKN3 expression in the endothelium is an important driver of processes, such as angiogenesis and metastasis formation. The i.v. administration of Atu027 via tail vein in mice bearing orthotopic pancreas (DanG cells) or prostate (PC-3 cells) xenografts led to significant reductions in both primary tumor growth and metastasis formation in lymph nodes. The authors showed that this effect was due to the knockdown of PKN3 expression and consequent changes in the tumor-associated lymphatic vasculature, which was suggested as evidence that Atu027 had antiangiogenetic effects. Atu027 was further tested in clinical trials. In Phase I clinical trial, Atu027 was well tolerated by patients with advanced refractory solid tumors. Stabilization of the disease was achieved in 14 of the 34 patients enrolled in the study. In a subsequent clinical study (Phase Ib/IIa) [31], Atu027 was tested in combination with gemcitabine for the treatment of locally advanced or metastatic pancreatic adenocarcinoma. The authors concluded that this treatment was safe and resulted in disease control in 7 out of 12 patients treated with a twice-weekly administration schedule.

Although cationic liposomes have been described as safe and effective transfecting agents in both preclinical and clinical studies, this is not a consensus. For example, one of the first cationic lipids to be used in transfecting liposomes, *N*-[1-(2,3-dioleyloxy)propyl]-*N*,*N*,*N*-trimethylammonium (DOTMA), has a high capacity of loading DNA and RNA, also being a highly efficient transfection agent, but it was shown to be nonspecific and prohibitively toxic [32].

One approach to increase the specificity of liposomes is to decorate their surface with PEG and target molecules. Zhuo et al. [33] described spherical, multilamellar PEGylated immunoliposomes with an average diameter of 122 nm. The surface of these vesicles was decorated with an antibody against endoglin (CD105), and the core was loaded with a plasmid carrying the mouse endostatin gene (pcDNA3.1-CSF1-endostatin). CD105 is a glycoprotein present on the plasma membrane expressed by endothelial cells, especially in those under rapid proliferation. The anti-CD105 mAb was used for both targeting tumor-associated endothelial cells and reducing angiogenesis. Endostatin, encoded by the therapeutic gene carried by the plasmid, would also reduce angiogenesis, as it is an antiangiogenetic peptide. This formulation was tested in an experimental model of human breast adenocarcinoma produced in NOD/SCID mice by the xenograft of MDA-MB-231 cells. The authors observed that these immunoliposomes accumulated in the tumors and were internalized by the target cells. In comparison to untargeted liposomes and the free plasmid, the immunoliposomes led to a significantly higher expression of endostatin in tumor-associated endothelial cells. The combined effect of anti-CD105 and endostatin expression reduced tumor growth by 71%.

In another study, Mu et al. [34] described an experimental gene therapy based on liposomes (Lipofectamine 2000^®^) decorated with apolipoprotein E (ApoE), a protein that has tumor-targeting properties. These liposomes were loaded with a plasmid (pcDNA3.1-pSurvivin-TK) containing the herpes simplex virus thymidine kinase/ganciclovir (HSVtk/GCV) suicide gene system driven by the promoter survivin. This system was proposed for the treatment of hepatocellular carcinoma, which is known usually to express receptors to LDL, thus being capable of binding ApoE. The gene therapy adopted by these authors is based on the conversion of innocuous nucleoside analogs, such as ganciclovir, into toxic metabolites by the HSVtk/GCV system. Tumor cells transfected with this system, and even adjacent cells would become sensitive to these drugs, as suggested by previous works [35,36]. The survivin promoter was chosen because hepatocellular carcinoma cells often express this protein at high levels. The in vivo model used to test the efficacy of this ApoE–liposome system was nude mice bearing xenografts of human hepatocellular carcinoma HepG2 cells. Tumor-bearing mice intravenously injected with the ApoE–liposomes loaded with pcDNA3.1-pSurvivin-TK, and later treated with ganciclovir, had a significantly longer survival period in comparison to the control groups. Interestingly, the decoration of the liposomes with ApoE improved the transfection of tumor cells with the therapeutic gene.

New cationic lipids have also been developed in an attempt to reduce toxicity and increase specificity. Zhao et al. [37] synthesized a series of tripeptide cationic lipids which the authors named CDL12, CDL14, CDO12, and CDO14. These lipids were composed of a cationic tripeptide—trilysine or triornithine—moiety, a carbamate linker, and two alkyl groups. These cationic lipids were mixed with the neutral lipid dioleoylphosphatidylethanolamine at a 1:1 ratio and used to prepare liposomes by the lipid film hydration method. These liposomes were able to transfer both plasmid DNA and siRNA into NCI-H460 and HEp-2 tumor cells in vitro. The lipid CDO14, composed of triornithine and C14 alkyl groups as tails, showed a better transfection efficiency in comparison to the other lipids. In vivo, in a lung tumor model established in BALB/c mice, the liposome produced with CDO14, injected intravenously, presented low toxicity and was able to transfect A549 cells with siRNA against c-Myc and VEGF.

The in vivo toxicity of CDO14-based liposomes, described in the previous paragraph, was further investigated by the same group and compared with the classical cationic lipid DOTAP—positive control [38]. The authors first evaluated the efficiency of siRNA delivery to tumors with a model of human lung carcinoma established in BALB/c mice by xenografts of A549 cells expressing firefly luciferase (Luc-A549). The mice were intravenously injected with different liposomes loaded with Luc-siRNA, which was supposed to suppress the expression of luciferase in Luc-A549 cells. The expression of luciferase was reduced by 70% in tumors of mice injected with CDO14 liposomes and was even lower than that observed with the positive control, DOTAP liposomes. Next, these two liposomes were used to transfect A549 tumor cells in vivo with type-1 insulin-like growth factor receptor siRNA (IGF-1R-siRNA). Although both liposomes reduced tumor growth, CDO14 liposomes were more efficient than DOTAP liposomes at suppressing tumors. Finally, the toxicity of CDO14 liposomes was lower than that of DOTAP liposomes, as evidenced in acute and subchronic toxicity tests. Additionally, we selected some examples of published articles that used liposomes to carry nucleotides in liposomes for the treatment of cancer (Table 1).

### 2.2. Immunostimulatory Effects

Another line of investigations in cancer therapy focuses on inducing immune responses against tumors with mRNA-based liposomal vaccines. Epitopes of tumor-specific or tumor-associated antigens are encoded by mRNAs and delivered to dendritic cells in vivo or ex vivo. Once delivered to the cytoplasm of the dendritic cell, the mRNA would be translated into peptides that could be presented by the major histocompatibility complex (MHC) system, inducing adaptive immune responses against tumor antigens [39]. This approach could be used alone or combined with different immunotherapy tools already available for clinical use, such as immune checkpoint inhibitors, that can fail to destroy tumors with low mutational burdens [40].

An example of this strategy can be found in the work by Kranz et al. [41]. Mice were injected i.v. with liposomal RNA-loaded vaccines, which delivered encoded antigens to antigen-presenting cells, such as macrophages and dendritic cells, in the spleen, lymph nodes, and bone marrow, inducing strong antigen-specific immune responses. This result was observed with encoded epitopes of different antigens, such as ovalbumin and influenza virus hemagglutinin A. Both effector and memory T-cell responses were triggered, and interferon-α-dependent rejection of tumors was observed in vivo. Thus, the authors suggested that this technology could be used to induce immunization against virtually any peptide antigen of interest for cancer immunotherapy.

Clinical studies have been conducted or are currently being developed with this technology for the treatment of cancers, such as pancreatic, ovarian, colorectal, and melanoma, as reviewed elsewhere [39]. For instance, FixVac (BNT111), an mRNA vaccine against melanoma, encodes four highly immunogenic, nonmutated, tumor-associated antigens that are commonly observed in melanoma, has been clinically tested alone or in combination with immune checkpoint PD1 blockade in phase 1 dose-escalation trial [41]. Both protocols induced durable objective responses in patients with unresectable melanoma. The clinical outcomes were accompanied by the induction of strong CD4^+^ and CD8^+^ T-cell responses. Of 50 treated patients, 39 patients (75%) exhibited CD4^+^ and CD8^+^ T-cell responses against at least one of the antigens encoded by the vaccine. Regarding adverse events, mild to moderate flu-like symptoms were observed. This vaccine is currently under evaluation in a phase 2 trial (NCT04526899) in patients with anti-PD-1-refractory or relapsed unresectable stage III and IV melanoma [39].

Another example of this strategy was proposed about two decades ago by Gursel et al. [42] in a study that tested the use of sterically stabilized cationic liposomes to improve the uptake and immunostimulatory activity of CpG motifs. These are bacterial-sourced unmethylated CpG nucleotides that stimulate mammalian immune systems and can act as immune adjuvants, immunoprotective agents, and antiallergens. The use of cationic liposomes approximately doubled the duration of induced immune protection against *Listeria monocytogenes* in a murine model when compared to free CpG nucleotides. More recently, Loira-Pastoriza et al. [43] showed that the encapsulation of an unmethylated CpG-rich nucleotide in cationic liposomes formed with dioleoyltrimethylammoniumpropane and dipalmitoylphosphatidylcholine enhanced antitumor activity following pulmonary delivery in a murine model of metastatic lung cancer. The authors showed that the nanosystem induced the production of a pro-apoptotic protein, cytokines, and chemokines.

Another reliable strategy involves the functionalization of nucleotides in the form of spherical nucleic acids onto the surface of liposomes. Guan et al. [44] proposed an immunostimulatory approach activated by spherical RNA selective for toll-like receptors (TLRs) 7/8 that are related to antiviral responses. This inventive structure consisted of a liposomal core functionalized with a dense shell of double-stranded RNA following insertion into the wall of the lipid core using hydrophobic cholesterol moieties. These RNA nanostructures potently activated TLR7/8 via the NF-κΒ signaling pathway, as evidenced using reporter cell lines and also by cytokine secretion and upregulation of other receptors. A similar nanostructure was developed by Callmann [45], aiming for triple-negative breast cancer treatment. The liposomal nanostructures consisted of immunostimulatory nucleotides as adjuvants co-delivered with tumor cell lysate as antigenic agents. The nanostructures were promising both in vitro and in vivo and in reducing tumor growth when compared to the controls consisting of simple mixtures of lysate and immunostimulatory nucleotides. Additionally, a list of articles that used liposomes to deliver nucleotides for immunostimulatory applications is presented in Table 2.

#### 2.2.1. Lipid-Based Ribozyme Delivery Systems for Theranostic Applications

Ribozymes are small RNA molecules or RNA–protein complexes. As their name indicates, they are enzymes of the ribonucleic acid type; these types of RNAs provide specific catalytic activity in biochemical reactions in species from all kingdoms of life, similar to protein enzymes [46,47,48,49]. Under appropriate conditions, these molecules exhibit specific catalytic activities in cis- and trans-cleavage or ligation of RNA and DNA and peptide bond formation [50]. Since divalent cations are necessary for the structural stabilization of RNA folding, and its catalytic mechanisms, unlike protein enzymes, which do not always require metals for their activity, ribozymes are always metalloenzymes [47,51].

In 1978 and during subsequent years, Sidney Altmann et al. [52] demonstrated that RNase P from *Escherichia coli*, an RNA–protein enzyme complex, had 5’-terminal RNA-cleaving activity involved in the processing of the precursor transfer RNA (tRNA), the term ribozyme was coined in 1982 by Thomas Cech et al. [53,54]. These authors discovered that a ribosomal RNA precursor sequence from the protozoan *Tetrahymena thermophila* cuts itself out of the mRNA it resides in and subsequently joins the two flanking ends [55]. This intramolecular reaction catalyzed by the same RNA, which is modified during reaction through a single turnover step, was considered the first proven catalytic activity of an RNA in the absence of any protein. They defined this reaction as self-splicing and, in 1989, Cech and Altmann shared the Nobel Prize in Chemistry for their work [50,56].

With the discovery of ribozymes, it was demonstrated that RNA can be both genetic material (like DNA) and a biological catalyst (like protein enzymes) and that it can catalyze chemical transformations in itself and other RNA molecules [47,55]. This dual-function feature of RNA contributed to suggesting that this biomolecule may have been important in the evolution of prebiotic self-replicating genome-limited organisms [55]. From 1982 until now, different natural ribozymes have been extensively characterized and classified based on their conserved sequences, secondary structures, and biochemical functions. The well-established natural ribozymes are classified regarding size into two large groups: (i) small ribozymes, such as the hairpin ribozyme, hammerhead ribozyme, Hepatitis delta virus (HDV) ribozyme, Varkud Satellite (VS) ribozyme, and glucosamine-6-phosphate riboswitch (glmS) ribozyme and (ii) large ribozymes that include group I and II of introns, the ribosome, spliceosome, and RNase P [57,58]. Small ribozymes are found in virus, virusoid, and satellite RNA genomes, in which they are responsible for the processing of rolling cycle replication intermediates to genome length [59,60]. In contrast, large ribozymes of group I and group II introns are abundant in organellar genomes, which catalyze two consecutive phosphotransesterification reactions, being crucial for the precise splicing of these introns and the maturation of their flanking exons [61,62]. RNase P participates in processes of the maturation of tRNA, using a hydrolysis reaction, and, in ribosomes, the ribozyme catalyzes a peptidyl transfer reaction in polypeptide synthesis [59,63].

Nonnative ribozymes can also be produced and characterized, such as the “leadzyme”, which is a small catalytic tRNA that catalyzes the cleavage of a specific phosphodiester bond in the presence of lead [64,65]. There are also DNAzymes, which were artificially developed based on DNA for various catalytic reactions [66]. Other artificial ribozymes catalyze reactions such as acylation and alkylation and the formation of glycosidic bonds, such as Diels–Alder ribozyme, which catalyzes the formation of carbon–carbon bonds between two small, non-nucleic acid substrates [67].

Since ribozymes catalyze site-specific cleavage reactions in vitro, in cell-free systems, and in living cells, these molecules are considered potential diagnostic and therapeutic tools by the cleavage of target RNAs [57,68,69], with some requirements that must be followed, regarding their role. Among these is the requirement that the ribozyme has to be conserved, has to be important to the disease or infection development, and has to be accessible, in other words, it has to be easy for the ribozyme to bind to the target region [70]. Among the advantages of their use is the fact that ribozymes have been reported as highly specific molecules regarding the recognition of their target and also as having low toxicity depending on the cell target. In contrast, the half-life of these molecules could be considered a disadvantage to their employment, and a robust delivery mechanism is very welcome and even needed to increase the robustness of their use. A delivery mechanism is also preferable regarding the integration of the ribozyme-encoding DNA sequence into the genome of the target organism, mainly because the exogenous delivery of ribozymes leads to the possibility that these molecules could be delivered directly to the site of their action [71].

#### 2.2.2. Therapeutic Applications by Ribozymes

Ribozyme-based therapeutic approaches have been developed for different purposes [72], such as anticancer [73,74], antiviral [75,76,77], and antiprion [78] treatments. The ribozymes in these potential therapeutic strategies are delivered to cells and tissues by two routes: exogenous and endogenous pathways. The exogenous pathway delivers synthesized ribozymes directly to tissues and cells as immediate-action therapeutic agents. Due to the low structural stability of RNA, the synthesized exogenous ribozymes are usually delivered as a chemically modified nucleotide ribozyme for its stabilization against nucleases degradation [79,80]. The endogenous pathway delivers the ribozymes in the form of its DNA-encoding sequence through viral gene therapy vectors [71,81].

In both delivery pathways, liposomes are commonly used for the liposome-mediated transfer of the DNA sequence encoding the ribozyme [82,83], in vitro transcribed ribozyme [84], or direct delivery of synthetic ribozymes [71,82,83]. Functional ribozymes are delivered either encapsulated within the aqueous liposome compartment or via lipid–nucleic acid complexes (lipoplexes) [27]. Different liposomal compositions have been studied for this purpose, and cationic lipids, in the form of liposomes or micelles, are the most effective carrier for delivery into the cells.

The inclusion of cationic lipids with a quaternary ammonium head group, which has permanent positive charges, in the composition of the liposomes provides a high affinity for negatively charged nucleic acid and cell membranes [85]. Usually, cationic lipids are used in combination with neutral or zwitterionic helper lipids, such as cholesterol, 1,2-dioleoyl-sn-glycero-3-phosphoethanolamine (DOPE), and polyethylene glycol (PEG)-lipid. These mixtures increase ribozyme transfection efficiency and the biocompatibility and biodegradability of cationic liposomes [86]. For example, the cationic lipid 1,2-di-*O*-octadecenyl-3-trimethylammonium-propane (DOTMA), or its biodegradable analogue 1,2-dioleoyl-3-trimethylammonium-propane (DOTAP), 2,3-dioleoyl-oxy-*N*-[2(spermine carboxamido) ethyl]-*N*,*N*-dimethyl-1-propaniminium trifluoroacetate (DOSPA), and (2S)-2,5-bis(3-aminopropylamino)-*N*-[2-(dioctadecylamino)acetyl] pentanamide (DOGS) in combination with DOPE are commercially available as lipoplex transfection agents (also called lipofectin, transfectins, lipofectamine, transfectam, or cytofectins) [71,87,88]. DOPE is an inverted cone-shaped lipid that aids cytosolic release by fusing cationic liposome and endosomal membranes, possibly through the formation of hexagonal lipid structures [89,90]. Consequently, the cationic lipid-liposomes with DOPE have the advantage of delivering the ribozyme to the cytoplasm via endosomal escape, without an intermediate endosomal stage, and therefore ribozymes would escape from the degradative lysosomal enzymes [91].

However, the positive head-group charge on cationic lipids can have cytotoxic effects, such as activation of pro-apoptotic and pro-inflammatory cell signaling pathways [92], and affects their circulation time and intracellular release efficiency [93,94]. Furthermore, the positive charges of cationic liposomes on their surface bind different plasma proteins and cause opsonization leading to phagocytosis [85]. Cytotoxic effects and nonspecific interactions can be reduced by controlling the compositions and proportions of lipids in the liposomes and by adding PEG and carboxymethyl-β-cyclodextrin to the surface of the cationic liposome [28,85,95].

Different studies have reported on the liposome-mediated delivery of exogenous ribozymes. In 1992, an exogenous ribozyme directed against the mRNA of the tumor necrosis factor α (TNFα) was delivered using lipofectin (DOTMA-DOPE mix) for cationic-liposome-mediated transfection into human promyelocytic leukemia cells (HL60) and peripheral blood mononuclear cells. TNFα plays an important role in inflammatory rheumatic diseases and modulates the expression of the class I antigens of the major histocompatibility complex and the cytokines interleukin 1 and interleukin 6 [96]. Using the same approach as lipopolysaccharide-induced TNFα gene expression inhibition, two hammerhead ribozymes (mRzl and mRz2) entrapped in cationic lipid liposomes were delivered to mice by intraperitoneal injection [97]. In both studies, the results showed lowered TNFα levels upon ribozyme treatment.

In 1993, another ribozyme, which mediates inhibition of leukemia bcr-abl gene expression in a Philadelphia chromosome-positive cell line, was delivered using the liposome of DOGS cationic lipid or transfectam [98]. In the same line of treatment as Philadelphia chromosome-positive chronic myelogenous leukemia, a multi-unit ribozyme transfection was used to reduce the level of bcr-ab1 mRNA. It was accomplished either by folate-receptor-mediated ribozyme transfection or DOTMA–DOPE liposomes as a carrier [84]. The authors showed that the liposome vector protected ribozymes in comparison to naked delivery; moreover, ribozyme uptake was more efficient by folate-receptor-mediated transfection.

Synthetic hammerhead ribozymes against the c-myb proto-oncogene, which is required for vascular smooth muscle cell proliferation, were delivered to rat aortic smooth muscle cells using lipofectamine or DOSPA–DOPE liposomes [99]. Using the same type of liposome, two modified hammerhead ribozymes, which mediate cleavage against the N-ras-mutant mRNA oncogene, were synthesized and evaluated with the N-ras/luciferase fusion gene as a reporter system in ex vivo HeLa cell assays [100]. A different approach used anionic and cationic liposomes fused with the hemagglutinating virus of Japan (HVJ) to deliver ribozyme-encoding plasmid into rat embryonic fibroblast cells. The authors showed that the uptake of ribozymes complexed with HVJ–cationic liposomes was 15–20 times greater compared with naked ribozymes and 4–5 times greater compared with ribozymes complexed with HVJ–anionic liposomes [82]. Most importantly, ribozymes complexed with HVJ–liposomes lead to a decrease in the target protein accumulation.

The modified PKCα ribozyme against malignant glioma growth, encapsulated in DOTAP liposomes, was injected into the center of the tumor and resulted in the inhibition of protein kinase Cα gene expression [101]. Ultimately, different cationic liposomes, such as lipofectamine (DOSPA–DOPE), lipofectin (DOTAP–DOPE), and 1,2-dimyristyloxypropyl-3-di-methyl-hydroxyethyl ammonium bromide (DMRIE)–DOPE, were also used to mediate the intracellular delivery of a fluorescein-labeled chimeric DNA–RNA ribozyme targeted to HIV-1 5’LTR [102]. In this study, it was clear that cationic-mediated ribozyme delivery needs further investigation, mainly regarding the specificity of its cytotoxicity effect and its effective quantity, before use as an in vivo therapeutic strategy.

#### 2.2.3. Diagnostic Applications by Ribozymes

Most studies report the use of ribozymes as therapeutic strategies, but their potential for diagnosis has also been discussed since the early 1990s. In patent WO 94/13833, a system to detect nucleic acids, proteins, or other molecules was described solely based on the ribozymes that would be enzymatically active in the presence of the target, leading to cleavage of a co-target and release of a detectably labeled molecule. In early 2000, a group of researchers from Sirna Therapeutics company, together with investigators from Thermo Electron Corp., Point of Care, and Rapid Diagnostics, developed a high-throughput assay with ribozymes that are able to detect the hepatitis C virus [103]. Upon binding to the target RNA, a half-ribozyme would go through a conformational change that allows substrate one with biotin on its 3′ and substrate two carrying a 5′-fluorescein to bind together. The product of this reaction could then be retained on a streptavidin-coated microtiter plate, and the fluorescein could be detected directly or its signal could be amplified by using an antifluorescein antibody conjugate with alkaline phosphatase enzyme. These ribozymes whose activity is regulated by external molecules are called allosteric ribozymes [103].

Ribozymes have also been indicated as a useful method for diagnosing a series of genetic disorders, which have been associated with triplet-repeat expansions. Penchovsky made two molecular input-AND-gate circuits based on designed allosteric hammerhead ribozymes, which undergo a conformational change to their active forms in the presence of two effectors, one of them is the RNA carrying the pathological number of repeats related to the disease [104]. In this study, as a proof of concept, the RNA carrying (GCG)11 repeats transcribed from the gene that encodes the human polyadenylate-binding nuclear protein 1 (PABPN1) were detected via ribozyme and differentiated from the nonrelated disorder RNA. The GCG repeat in the PABPN1 gene is related to the oculopharyngeal muscular dystrophy (OPMD) disorder. The system was expanded even more from three input-AND-gate circuits to a multiplex decoder circuit. Although it was a great advance toward the use of ribozymes as a diagnostic method of repeat expansions related to several genetic disorders, making it possible to expand to other diseases, the authors failed to show the system’s functionality to detect a specific RNA among the whole range of endogenous molecules sampled from patients [104].

Ribozymes could also be used together with other molecules, such as aptamers, for diagnostic purposes. For example, Liao et al. developed a colorimetric assay using an aptamer, an allosteric hammerhead ribozyme, an isothermal exponential amplification reaction (EXPAR), and the peroxidase activity on 3,3′,5,5′-tetramethylbenzidine (TMB) substrate as a detection pathway. Although the bronchodilator Theophylline was used as an analyte in this study, it is possible to foresee the application of this diagnostic method to molecules related to diseases and viral infections [105].

Although diagnostic methods have been discussed as in vitro assays, it is of interest to develop tools for in vivo diagnosis, and, for that, delivery mechanisms are of essential importance. Cationic lipids have been successful in facilitating ribozyme delivery since the early 1990s and 1998 [102]; as discussed previously in this review, they have been applied in therapeutic strategies with these biomolecules, which might pave the way for their potential use in vivo diagnosis. Moreover, Table 3 brings some examples of articles that used ribozymes in association with liposomes.

### 2.3. Veterinary Applications

For decades, the potential use of liposomes-loading nucleotides in the veterinary field to treat infectious diseases and for therapeutic vaccines was considered a challenging issue, since emerging technologies for veterinary medicine must be cost-effective, show high efficiency, present large-scale feasibility, and be safe for livestock and domestic animals. Although nucleotides entrapped in liposomes could not match some of these criteria, recent advances strongly suggest the potential use of these nanostructures in modern veterinary medicine.

Recently, Camussone et al. [106] successfully evaluated the ability of a subunit vaccine composed of four recombinant molecules formulated with cationic liposomes and CpG oligodeoxynucleotides to confer protection against *Staphylococcus aureus* intra-mammary infections in heifers and cows. Cationic liposomes and CpG oligodeoxynucleotides were also used as adjuvants to improve the immunogenicity of a truncated recombinant protein-based vaccine targeting *Neospora caninum* in cattle, and it proved also to enhance the humoral immune response after challenge [107].

DNA vaccines have been tested for the treatment of avian diseases with the potential to be used in the poultry industry. An example is a study developed by Liu et al. [108]. Chicken infectious anemia (CIA) is an immunosuppressive disease caused by the chicken anemia virus that poses relevance significant threat to the poultry industry worldwide.

Mueller et al. [109] performed a pilot study aiming to evaluate the immunostimulatory effects of cationic liposome–plasmid DNA complexes associated with some selected allergen extracts applied intradermally for the treatment of canine atopic dermatitis after the failure of conventional therapies. Their results indicated that this type of liposome–nucleic acid complex vaccine may allow the use of lower antigen doses for the treatment of this disease in dogs.

*Staphylococcus aureus* biofilm–based infections using conventional antibiotics are a challenging issue since only sublethal doses of the biofilm can be administered to both humans and animals. Recently, a study conducted by Ommen et al. [110] presented a tailored DNA aptamer-targeted liposomal nanosystem for binding and specific delivery of conventional antibiotics locally in *S. aureus* biofilms. This delivery system was able to eradicate *S. aureus* biofilm after 24 h of in vitro treatment and highlighted the potential of this strategy for in vivo treatment of *S. aureus* biofilm infections.

Some animals are widely considered biofactories for protein-based biopharmaceuticals. In this context, rabbits are commonly used in polyclonal antibody production. Since increased antibody concentrations can be yielded with the use of nano-adjuvants associated with antigens, Lee et al. [111] evaluated cationic liposome–nucleotide complexes as adjuvants for polyclonal antibody production in New Zealand white rabbits. They demonstrated that cationic liposome–nucleotide complexes could be promising adjuvants for high-yielding polyclonal antibody production in rabbits, similar to other common adjuvants, but without developing the undesired skin lesions that are typical of conventional compounds. Additionally, we prepared Table 4 with some examples of the use of nucleotides entrapped in liposomes for veterinarian applications.

### 2.4. Nucleic Acids and Liposomes in the Prevention of Neglected Tropical Diseases

As mentioned earlier, the chaotic health scenario imposed by the COVID-19 pandemic accelerated the progress of nucleotide-based prophylactic vaccine development for infectious diseases. In a paradigm shift, prophylactic vaccines containing the SARS-CoV-2 virus spike surface protein gene delivered in both DNA and mRNA forms were approved in record time for emergency use [112,113]. The use of these methodologies has brought about several ethical questions regarding both their use and the speed of their approval [114,115]. However, the approach of nucleic acids as a vaccine for infectious diseases is not a novelty unique to COVID-19, since, before 2020, the use of DNA for developing therapies or prophylaxes for humans was addressed [116], such as vaccines for dengue [117], hepatitis [118], malaria [119], Chagas disease [120], and schistosomiasis [121]. RNA use began as antitumor therapy [122] and is currently expanding to a wide range of diseases including neglected diseases.

Neglected diseases or neglected tropical diseases (NTDs) constitute a group of 20 conditions, including dengue, chikungunya, trypanosomiasis, and leishmaniasis. These diseases primarily affect impoverished communities and disproportionately affect women and children, culminating in devastating health, social and economic consequences for billions of people [123,124].

In the last 3 years, research around the development of prophylactic strategies for NTDs has increased on an exponential scale thanks to the new technologies approved for COVID-19 vaccines. Nucleic-acid-based vaccines, including viral vectors, plasmid DNA, and mRNA, are suitable for rapid response applications due to their ability to induce robust immune responses besides their potential for fast production, flexibility, and reduced ultimate cost [125]. Moreover, following vaccination, nucleic-acid-based vaccines mimic a virus infection as they result in the expression of vaccine antigens in situ, inducing a humoral and cytotoxic T-cell response [126].

The emergence of cases and climate change shows just how much a concerted effort on a global scale to develop safe and effective vaccines to prevent NTD arboviruses, such as dengue and zika, is needed. Wollner et al. [127] developed a vaccine against DENV serotype 1 with mRNA-LNP encoding wild-type prM and E proteins and a second construct (ΔFL) without the E protein fusion loop epitope involved in ADE (antibody-dependent enhancement) of infection. All mice from WT and ΔFL vaccine cohort seroconverted and elicited humoral and cell-mediated immunity following a two-dose vaccination regimen, with neutralizing titers similar to or higher than other published vaccine prototypes to DENV1. In addition, the DENV1 prM/E mRNA-LNP vaccine protects against a lethal DENV1 challenge in an immunocompromised AG129 mouse. Regarding ADE potential, DENV1 prM/E mRNA vaccines did not induce cross-reactive antibodies that elicit heterotypic enhancement, in contrast to a viral infection.

Another viral NTD subjected to prevention using nucleic acid is the zika. The zika virus has been identified as the etiologic agent of microcephaly in newborns and Guillain–Barré syndrome in adults [128,129]. Several immunization platforms are currently under study for a ZIKV vaccine, including plasmid DNA, inactivated virus particles, protein subunits, and adenovirus-based vectors. Currently, in the phase II study, the experimental mRNA-1893 zika virus vaccine has shown promising results as in previous studies [130]. The mRNA-1893 vaccine was formulated in ionizable lipid-based LNPs and generated neutralizing antibodies 1/20th of the dose and provided complete protection against ZIKV challenge in nonhuman primates. In addition, prime and boost vaccination with mRNA-1893 10 µg produced high antibody titers. Furthermore, no animal immunized with mRNA-1893 showed detectable viral load after the challenge.

As in most viral infections, the elimination of intracellular parasites and the infection rely on potent cellular and humoral immune responses. This is the case for intracellular pathogens, such as *Trypanosoma cruzi*, which causes Chagas disease, and *Leishmania* sp., which causes leishmaniasis. Effective immunity against these parasites in mammalian models depends on the triggering of a Th1-type response, characterized by the production of high levels of IFN-γ, IL-12, and IL-2, among other pro-inflammatory cytokines [131]. Nucleic acid vaccines have intrinsic adjuvant properties due to their recognition by specific pattern recognition receptors (PRRs) and elicitation of innate immune responses, which are critical for the maturation of dendritic cells (DCs) to enhance induction of subsequent adaptive immune responses [132].

In most approaches involving DNA vaccines so far, the main obstacle is low efficacy. In the study targeting the F subunit of the *Leishmania tropica* V-ATPase protein [133], the authors inoculated three doses of the vaccine formulated with 100 μg of plasmid DNA containing the V-ATPase subunit F gene, at 2-week intervals, into female BALB/c mice. Following vaccination, 41.9% of the animals showed a reduction in *L. tropica* infection in addition to induction of a mixed Th1 and Th2 cell response. The authors suggest that adjuvant use is needed to further stimulate the Th1 response. However, several studies have shown that the delivery of naked genetic material, i.e., without polymeric or lipidic protection, reduces the chance of success in capturing DNA [134].

For Chagas disease, DNA-based vaccines have been developed and delivered with different coadjuvants [135,136]. DNA vaccines may provide an alternative for both the prevention and treatment of a variety of diseases, including Chagas disease. Therefore, it is suggested that suitable DNA vectors encoding antigens could be encapsulated in cationic lipid nanoparticles and used as vaccine systems to deliver an efficient vaccine against *T. cruzi* [137].

Nanocarriers have been proven to facilitate the delivery of genetic material to antigen-presenting cells, such as cationic lipids, which are among the most used for the delivery of messenger RNA in genetic material vaccines [134]. However, when it comes to research with NTDs, there is little investment in research involving nano-encapsulated genetic material. Even after the growing wave of the use of mRNA and DNA as a prophylactic strategy for various diseases, the field of neglected diseases remains lagging and is thus in need of investment in cutting-edge research. A list with some examples of nucleotides entrapped in liposomes for NTD application is presented in Table 5. 

## 3. Conclusions

As discussed previously, the use of liposomes to protect and deliver nucleotides for biomedical applications is not new and has been a useful strategy in the last 30–40 years. However, the technology of liposomes containing nucleotides was only widely applied during the COVID-19 pandemic, for the application of mRNA vaccines. After this first large application in human beings, other initiatives aiming at different types of diseases have been proposed in the literature. As presented in this review, we will probably face an important revolution in terms of therapeutic and diagnostic use of these nucleotides for biomedical applications. Thus, a variety of solutions that rely on nucleotides entrapped in cationic liposomes will be created and offered to the public in the next years.

## 4. Perspectives for Biomedical Applications

In addition to these technical perspectives, the development of these technologies using nucleotides as therapeutic and/or diagnostic molecules has a great chance of completely revolutionizing the treatment and even the control of several diseases of biomedical interest, as presented in this review. In terms of historical perspectives, the incorporation of these technologies can be ranked as one of the greatest advances in the biomedical area of the century. Perhaps, we can compare it to the development of antibiotics, which is considered one of the greatest advances in medicine in the last century. For sure, the COVID-19 pandemic was the big challenge faced by these technologies; however, other more important issues, such as cancer, neglected diseases, and food production, may also be benefited from this technological revolution.

## Figures and Tables

**Figure 1 pharmaceutics-15-00873-f001:**
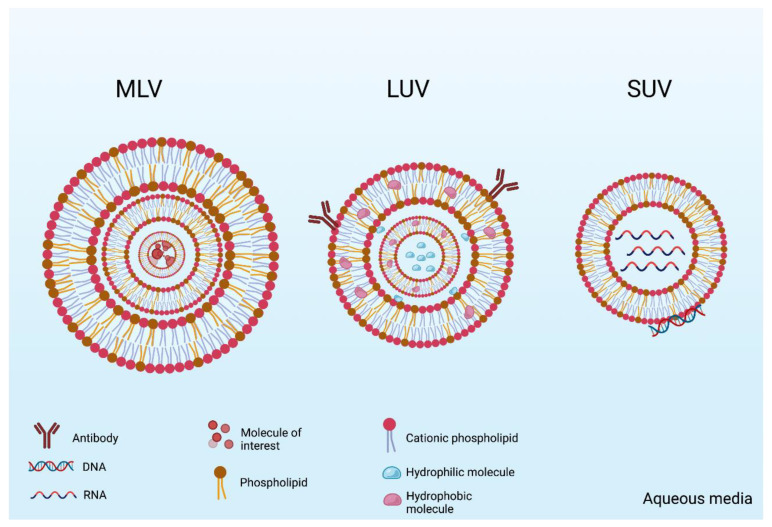
A schematic representation of the different types of liposomes: multilamellar vesicles (MLVs), large unilamellar vesicles (LUVs), and small unilamellar vesicles (SUVs). The vesicles can be decorated with surface molecules such as antibodies and can encapsulate both hydrophilic and hydrophobic molecules. Nucleic acids usually are entrapped when the liposome phospholipids have cationic charges. For nucleotide liposomes, the DNA or RNA molecule can be entrapped in the inner aqueous space or be absorbed on the vesicle surface due to the cationic phospholipids charges.

**Figure 2 pharmaceutics-15-00873-f002:**
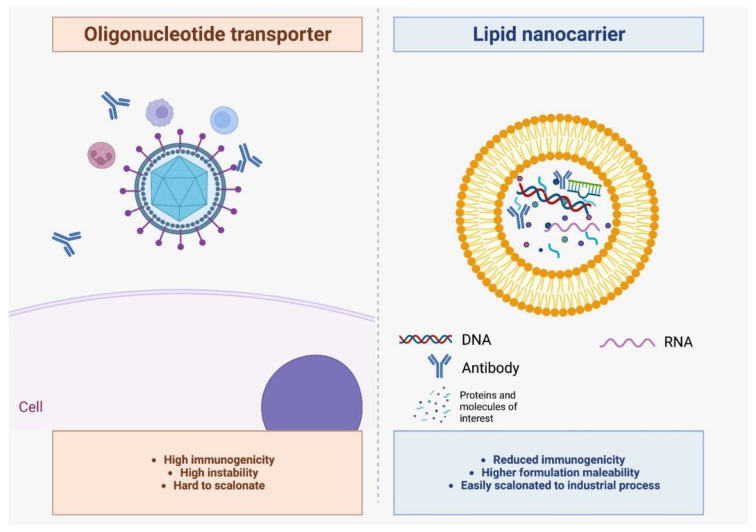
Different types of nucleic acid transporters. In comparison to liposomes, viral vectors are more immunogenic due to the presence of viral molecules that can be recognized by our immune system. On the other hand, liposomes, prepared with phospholipids, are well tolerated by immune cells. Moreover, the surface of the liposomes can be decorated with target molecules, such as antibodies or other biomolecule motifs, to deliver the vesicles to specific tissue sites.

**Figure 3 pharmaceutics-15-00873-f003:**
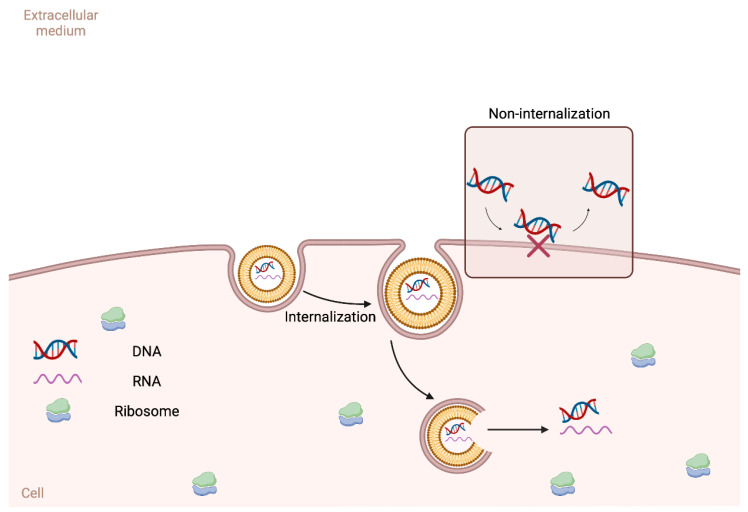
A representative scheme describing how the nucleotides entrapped in liposomes are internalized by cells. As hydrophilic acids, the transport of DNA/RNA molecules is not so easy. The association of the nucleotides with liposomes may improve this internalization process.

**Figure 4 pharmaceutics-15-00873-f004:**
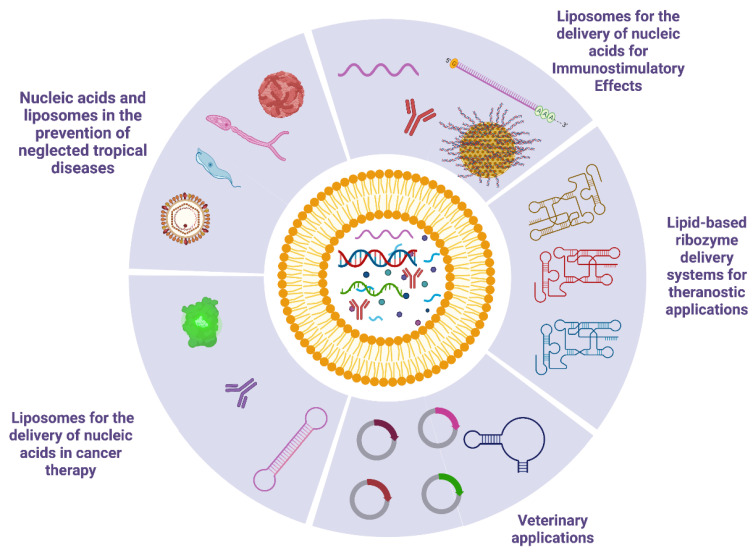
Examples of different nucleotides and their biomedical applications described in this review.

**Table 1 pharmaceutics-15-00873-t001:** Liposomes for the delivery of nucleic acids in cancer therapy.

Reference	Nucleotide Carried	Target Disease	Main Outcome
Aleku et al., 2008 [30]	siRNA (Atu027), targets the expression of protein kinase N3 (PKN3)	Preclinical—prostate and pancreatic cancer	Reduction in tumor vessels density
Schultheis et al., 2016 [31]	siRNA (Atu027), targets the expression of protein kinase N3 (PKN3)	Clinical trial—advanced pancreatic carcinoma	Atu027, in combination with gemcitabine, was well tolerated in the clinical trial. The results suggest the efficacy of Atu027 in the pancreatic carcinoma therapy
Zhuo et al., 2018 [33]	pcDNA3.1-CSF1-endostatin, mouse endostatin gene	Preclinical—mouse/breast cancer	Tumor suppression by 71%
Mu et al. (2020) [34]	Herpes simplex virus thymidine kinase/ganciclovir (HSVtk/GCV) suicide gene system	Preclinical—mice hepatocellular carcinoma	Suppressed tumor growth and extended mice survival time
Zhao et al. (2015) [37]	siRNA, for silencing c-Myc and VEGF oncogenic pathways	Preclinical—mice lung cancer	siRNA silenced distinct oncogenic pathways (c-MYC and VEGF) in mice lung tumors
Zhu et al. (2019) [38]	siRNA, IGF-1R-siRNA to inhibit tumor growth	Preclinical—mice lung cancer	Controlled lung tumor growth tumor-bearing mice

**Table 2 pharmaceutics-15-00873-t002:** Liposomes for the delivery of nucleic acids for immunostimulatory effects.

Reference	Nucleotide Carried	Target/Application	Main Outcome
Kranz et al. (2016) [41]	RNA, RNA-Lipoplex that triggers interferon-α (IFNα) release	Macrophages and dendritic cells	Effector and memory T-cell responses were triggered, and interferon-α-dependent rejection of tumors was observed in vivo
Lorentzen et al. (2022) [39]	mRNA—review article with different vaccine applications	Vaccine against melanoma	Induced durable objective responses in patients with unresectable melanoma
Gursel et al. (2001) [42]	CpG oligonucleotides, unmethylated oligodeoxynucleotides containing CpG motifs like bacterial nucleotides that trigger immune cells	Immune adjuvants, antiallergens, and immunoprotective agents	Increased 15- to 40-fold antigen immunization in mice models
Loira-Pastoriza et al. (2021) [43]	CpG oligonucleotides, unmethylated oligodeoxynucleotides containing CpG motifs similar to bacterial nucleotides that trigger immune cells	Immunoadjuvant for murine lung tumors	Liposomes increased CpG effectiveness in controlling murine lung tumors
Guan et al. (2018) [44]	Spherical nucleic acids, RNA selective for targeting toll-like receptors (TLRs) 7/8	Dendritic cells	Potently activate TLR7/8 via NF-κΒ signaling
Callmann (2020) [45]	Spherical nucleic acids, which are oligonucleotides functionalized with hydrophobic carbon chains to increase their liposome encapsulation. In this report, authors used immunostimulatory oligonucleotides	Preclinical—murine breast cancer	Reduce both primary tumor and metastatic growth due to immunostimulant effects

**Table 3 pharmaceutics-15-00873-t003:** Lipid-based ribozyme delivery systems for theranostic applications.

Reference	Nucleotide Carried	Target/Application	Main Outcome
Sioud et al., 1992 [96]	Hammerhead ribozyme-mediated suppression of TNF-α through transfection with cationic liposomes.	HL60 and PBMNC cell lines in models of inflammatory rheumatic diseases.	90% and 85% reduction in tumor necrosis factor alpha mRNA and protein, respectively.
Sioud, 1996 [97]	Suppression of LPS-induced TNF-α through hammerhead ribozyme-mediated transfection with cationic liposomes.	Mice/Inflammatory rheumatic disease.	50–70% inhibition of TNF-α gene expression through ribozymes.
Snyder et al., 1993 [98]	Hammerhead ribozymes with RNA and RNA–DNA hybrid structures. Ribozyme structure cleaved bcr-abl (abnormal fused gene present in chronic leukemia) mRNA.	EM-2 cell line and patient-derived cell lines with blast crisis CML.	Reduced levels of *bcr-abl* mRNA involved in the pathogenesis of Ph^1^+ leukemia.
Leopold et al., 1995 [84]	Transfection of double- and triple-unit ribozymes using liposomes or folic acid–polylysine as carriers.	32D cells from murine myeloblasts associated with CML model disease.	Reduced levels of *bcr-abl* mRNA from the tyrosine kinase fusion gene resulting from the *bcr* gene on chromosome 22 and *abl* gene on chromosome 9.
Jarvis et al., 1996 [99]	Hammerhead ribozymes complexed with DOSPA. Cleave c-myb RNA.	Rat aortic smooth muscle cells from female rats.	Reduction in *c-myb* proto-oncogene mRNA associated with the proliferation of vascular smooth muscle cells.
Scherr et al., 1997 [100]	Hammerhead ribozymes complexed with DOSPA. Targeting N-ras oncogene.	HeLa cells assays.	Cleavage of *N-ras* oncogene RNAm in HeLa cells and reduced expression of *N-ras/luciferase* fusion gene.
Sioud and Sørensen, 1998 [101]	The complex of modified PKCα ribozyme with cationic liposome. Inhibition of protein kinase Cα.	Solid tumors of malignant glioma.	Inhibition of PKCα gene expression.
Kossen et al., 2004 [103]	Allosteric half-ribozyme used to detect natural viral sequences variants.	Detect the hepatitis C virus.	Upon binding to the target RNA, the half-ribozyme would go through a conformational change that allows substrate one with biotin on its 3′ and substrate two carrying a 5′-fluorescein to bind together. Being directly detected or its signal could be amplified.
Penchovsky, 2012 [104]	Allosteric hammerhead ribozyme.	Detect oculopharyngeal muscular dystrophy (OPMD) disorder.	Allosteric hammerhead ribozymes that undergo a conformational change to their active forms in the presence of the RNA-carrying (GCG)_11_ repeats transcribed from the gene that encodes the human polyadenylate-binding nuclear protein 1 (PABPN1).
Liao et al., 2018 [105]	Aptamer and an allosteric hammerhead ribozyme.	Detect the bronchodilator theophylline.	Upon binding of the ligand-target and the aptamer, the ribozyme under a conformational change and self-cleavage triggers the amplification of a reporter, which oxidizes a substrate leading to a visible color change.

**Table 4 pharmaceutics-15-00873-t004:** Veterinary applications.

Reference	Nucleotide Carried	Target/Application	Main Outcome
Camussone et al. (2022) [106]	α-toxin gene (plasmid). CpG oligonucleotides, unmethylated oligodeoxynucleotides containing CpG motifs like bacterial nucleotides that trigger immune cells	Immunization against *S. aureus* intramammary infection (cow mastitis)	Reduction, but not significant of animal mastitis
Liu et al. (2022) [108]	DNA plasmid (infectious anemia virus sequences) The plasmid was used as a vaccine booster in combination with the antigen exposition	Chicken viral infectious anemia	DNA vaccine alone did not protect against the infection. The combined vaccine (plasmid + antigen) had the best protection against the infection
Mueller et al. (2005) [109]	DNA plasmid containing CpG oligonucleotides, which are unmethylated oligodeoxynucleotides that trigger immune cells	Immunostimulant against refractory canine atopic dermatitis	The vaccine decreased some signs and symptoms related to atopic dermatitis, as well as a reduction in IL-4 production
Ommen et al. (2022) [110]	DNA aptamers that target and bound to *S. aureus* cells	The DNA aptamers were used as targeting molecules for liposome delivery to *S. aureus* biofilms	The aptamer was useful to improve the delivery of antibiotics entrapped in nano-sized liposomes
Lee et al. (2017) [111]	Proprietary oligonucleotides (F5881, F5506, T2684, Sigma-Aldrich, St Louis, MO, USA)	Adjuvants in polyclonal antibody production in rabbits	The liposome oligonucleotide complexes were effective as immune adjuvants for polyclonal antibody production in rabbits

**Table 5 pharmaceutics-15-00873-t005:** Nucleic acids and liposomes in the prevention of neglected tropical diseases.

Reference	Nucleotide Carried	Target/Application	Main Outcome
Yin et al., 2021 [118]	Hepatitis B virus plasmids expressing HBcAg and HBeAg	Hepatitis B (vaccine)	Strong Th1 and Th2 immune response resulting in the elimination of the virus after the challenge
Rodríguez-Morales et al., 2012 [120]	Plasmids containing *Trypanosoma cruzi* genes encoding the Tc SP (trans-sialidase protein) and Tc SSP4 expression (amastigote-specific protein)	Chagas disease (vaccine)	Induction of moderate level of protection in immunized dogs
Li et al., 2011 [121]	Plasmid containing the gene encoding glutathione S-transferase of *Schistosoma japonicum*	Schistosomiasis (vaccine)	The reduction rate of worm and egg burdens in the pEGFP-Sj26GST plus CIM group was more than 68% and higher than that in pEGFP-Sj26GST alone (*p* < 0.01)
Wollner et al., 2021 [127]	Nucleotide-modified mRNA vaccine encoding the membrane and envelope structural proteins from DENV serotype 1 encapsulated in lipid nanoparticles	Dengue (vaccine)	Robust antiviral immune responses comparable to viral infection, with high levels of neutralizing antibody titers and antiviral CD4^+^ and CD8^+^ T cells
Bollman et al., 2022 [130]	mRNA of VLPs Zika virus. mRNA-1325 encodes Zika Virus membrane envelope proteins	Zika (vaccine)	Complete protection against ZIKV challenge in nonhuman primates

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
