# Peer review of "Nucleotides Entrapped in Liposome Nanovesicles as Tools for Therapeutic and Diagnostic Use in Biomedical Applications"

_pharmaceutics, 2023, doi:10.3390/pharmaceutics15030873_

Round 1

Reviewer 1 Report

The manuscript written by Cardador et al. is about summarizing the use of liposomes both in therapy and diagnostics. The manuscript is very extensive, well-organized, and well-written. The topic and quality of this manuscript is appropriate to be published in Pharmaceutics. However, the authors should consider the following suggestions to be included in the manuscript before being accepted.

Minor comments

(1) The type of this manuscript should contain tables and thus complement the main text included in the article. For example, the authors may include in each section a summary of liposomes that have been described including the most significant outcomes.

(2) I suggest including additional figures of some examples selected by the authors.

(3) Besides conclusions, the authors should include another section describing a personal overview and future perspectives of the field.

Author Response

Review 1

  1. The type of this manuscript should contain tables and thus complement the main text included in the article. For example, the authors may include in each section a summary of liposomes that have been described including the most significant outcomes.

Authors Response:

Thanks for your suggestion. The inclusion of summarized tables may improve the reader's understatement of the different outcomes of each application presented in the review. To clarify this point, we include 5 tables, one for each application described in the article. The tables were inserted in the revised manuscript in the followings lines:

Line 370: Table 1: Liposomes for the delivery of nucleic acids in cancer therapy;

Line 424: Table 2: Liposomes for the delivery of nucleic acids for Immunostimulatory Effects;

Line 595: Table 3: Lipid-based ribozyme delivery systems for theranostic applications;

Line 632: Table 4: Veterinary application;

Line 705: Table 5: Nucleic acids and liposomes in the prevention of neglected tropical diseases.

               For each table, we included the following points:

Reference

Nucleotide Carried

Target/Application 

Main Outcome

  1. I suggest including additional figures of some examples selected by the authors.

Authors Response:

Thanks for your suggestion. Regarding the figures presented, we revised both original figures and include two additional figures. For Figures 1 and 2 we include some additional information to improve the concept understandment. For the new Figures 3 and 4, we present how the liposome can improve the nucleotide delivery for cells; and which are the main applications presented in the review paper. Figure legends are presented in the following lines:

Lines 50: Figure 1: Schematic representation of the different types of liposomes. multilamellar vesicles (MLV); large unilamellar vesicles (LUV); and small unilamellar vesicles (SUV). The vesicles can be decorated with surface molecules such as antibodies and can encapsulate both hydrophilic and hydrophobic molecules. Nucleic acids usually are entrapped when the liposome phospholipids have cationic charges.

Lines 94: Figure 2: Different types of nucleic acid transporters. In comparison to liposomes, viral vectors are more immunogenic due to the presence of viral molecules that can be recognized by our immune system. On the other hand, liposomes, prepared with phospholipids are well tolerated by immune cells.

Lines 169: Figure 3: Representative scheme describing how the nucleotides entrapped in liposomes are internalized by cells. As hydrophilic acids, the transport of DNA/RNA molecules is not so easy. The association of the nucleotides with liposomes may improve this internalization process.

Lines 277: Figure 4: Examples of different nucleotides and their biomedical applications described in this review.

We also included the following sequence to describe figure 2 (lines 97-98): “Moreover, the liposomes surface can be decorated with target molecules such as antibodies or other biomolecules motifs to deliver the vesicles to specific tissue sites.” Figure 3 (lines 163-167): “For instance, nucleotide molecules are not well delivered to cells due to their high hydrophilicity. The encapsulation of these molecules in liposomes can improve this delivery, allowing them to reach the inner compartments of live cells. In Figure 3, we present a representative scheme with this process. While free molecules are not well absorbed, nucleotides entrapped in liposomes can pass throw the membrane more easily.”

  1. Besides conclusions, the authors should include another section describing a personal overview and future perspectives of the field.

Authors Response:

Thanks for your suggestion. We include a perspective section with the personal author's view of the field. The idea was to share how we observe the evolution of these technologies for the applications that were discussed during the review. We included the following sentence in the lines 717-726:

Perspectives for Biomedical Applications

In addition to these technical perspectives, the development of these technologies using nucleotides as therapeutic and/or diagnostic molecules has a great chance of completely revolutionizing the treatment and even the control of several diseases of biomedical interest, as presented during the present review. In terms of historical perspectives, the incorporation of these technologies can be ranked as one of the greatest advances in the biomedical area of the century. Perhaps, we can compare it to the development of antibiotics, which is considered one of the greatest advances in medicine in the last century. For sure, the COVID-19 pandemic was the big challenge faced by these technologies, however, other super important issues, such as cancer, neglected diseases, and food production may also be benefited from this technological revolution.

Reviewer 2 Report

The manuscript entitled "Nucleotides entrapped in Liposome Nanovesicles as tools for therapeutic and diagnostic use in biomedical applications" by Camila Magalhães Cardador et al.

In this article, although the topic might be interesting, the authors failed to deliver the central theme which limited the enthusiasm toward this work.

- The article is rather short and not comprehensive at all.

- The preparation of figures are poor that there is no key or explantion of the cartoon/schematics which makes it difficult to follow.

- Figure 1, the drawings among panels are not in real scale which make readers difficult to comprehend the differences among these vesicles, as in general the LUV should be the biggest in size as compared to the other two.

- Figure 2, left panel, it is unclear why it is called oligonucleotide transporter and where is the oligonucleotides? Right panel, similarly, the drawing is confusing too as if only DNA can be entrapped. As a matter of fact, various types of natural or artificial made biomolecules can be entrapped.

- Only two simple figures were presented with little useful information in which no new knowledge can be learnt.

- Examples of ultilization of these vesicles should be summarized as a table for the ease of reading.

- Typos and unfriendly mode of English usage can be found.

Author Response

Reviewer 2

  1. The article is rather short and not comprehensive at all.

Authors Response:

Thanks for your suggestion. We understand that the use of nucleotides could revolutionize biomedical applications in the coming years. The applications are myriad, perhaps including some that weren't even planned at the time. In this sense, our review is not intended to include all these possible applications, but those that we consider the most relevant from our point of view. Perhaps, the absence of tables summarizing these main results was a flaw in the presentation of the first version of the article. To improve this point, we have included 5 tables summarizing the main results presented throughout the review. These tables were included in the following lines of the revised version of the article:

Line 370: Table 1: Liposomes for the delivery of nucleic acids in cancer therapy;

Line 424: Table 2: Liposomes for the delivery of nucleic acids for Immunostimulatory Effects;

Line 595: Table 3: Lipid-based ribozyme delivery systems for theranostic applications;

Line 632: Table 4: Veterinary application;

Line 705: Table 5: Nucleic acids and liposomes in the prevention of neglected tropical diseases.

               For each table, we included the following points:

Reference

Nucleotide Carried

Target/Application 

Main Outcome

We also included two additional Figures (3 and 4) and revised the original Figures 1 and 2.

  1. The preparation of figures are poor that there is no key or explantion of the cartoon/schematics which makes it difficult to follow.

Authors Response:

Thanks for this suggestion. To improve this point, we revised the two original figures, including additional information on both of them. For Figures 1 and 2 we include some additional information to improve the concept understandment. For the new Figures 3 and 4, we present how the liposome can improve the nucleotide delivery for cells; and which are the main applications presented in the review paper. Figure legends are presented in the following lines:

Lines 50: Figure 1: Schematic representation of the different types of liposomes. multilamellar vesicles (MLV); large unilamellar vesicles (LUV); and small unilamellar vesicles (SUV). The vesicles can be decorated with surface molecules such as antibodies and can encapsulate both hydrophilic and hydrophobic molecules. Nucleic acids usually are entrapped when the liposome phospholipids have cationic charges.

Lines 94: Figure 2: Different types of nucleic acid transporters. In comparison to liposomes, viral vectors are more immunogenic due to the presence of viral molecules that can be recognized by our immune system. On the other hand, liposomes, prepared with phospholipids are well tolerated by immune cells.

Lines 169: Figure 3: Representative scheme describing how the nucleotides entrapped in liposomes are internalized by cells. As hydrophilic acids, the transport of DNA/RNA molecules is not so easy. The association of the nucleotides with liposomes may improve this internalization process.

Lines 277: Figure 4: Examples of different nucleotides and their biomedical applications described in this review.

We also included the following sequence to describe figure 2 (lines 97-98): “Moreover, the liposomes surface can be decorated with target molecules such as antibodies or other biomolecules motifs to deliver the vesicles to specific tissue sites.” Figure 3 (lines 163-167): “For instance, nucleotide molecules are not well delivered to cells due to their high hydrophilicity. The encapsulation of these molecules in liposomes can improve this delivery, allowing them to reach the inner compartments of live cells. In Figure 3, we present a representative scheme with this process. While free molecules are not well absorbed, nucleotides entrapped in liposomes can pass throw the membrane more easily.”

  1. Figure 1, the drawings among panels are not in real scale which make readers difficult to comprehend the differences among these vesicles, as in general the LUV should be the biggest in size as compared to the other two.

Authors Response:

Thansks for your suggestion. We updated Figure 1 for the revised article version. Lines 50: Figure 1: Schematic representation of the different types of liposomes. multilamellar vesicles (MLV); large unilamellar vesicles (LUV); and small unilamellar vesicles (SUV). The vesicles can be decorated with surface molecules such as antibodies and can encapsulate both hydrophilic and hydrophobic molecules. Nucleic acids usually are entrapped when the liposome phospholipids have cationic charges.

  1. Figure 2, left panel, it is unclear why it is called oligonucleotide transporter and where is the oligonucleotides? Right panel, similarly, the drawing is confusing too as if only DNA can be entrapped. As a matter of fact, various types of natural or artificial made biomolecules can be entrapped.

Authors Response:

Thansks for your suggestion. We updated Figure 2 for the revised article version. Lines 94: Figure 2: Different types of nucleic acid transporters. In comparison to liposomes, viral vectors are more immunogenic due to the presence of viral molecules that can be recognized by our immune system. On the other hand, liposomes, prepared with phospholipids are well tolerated by immune cells.

  1. Only two simple figures were presented with little useful information in which no new knowledge can be learnt.

Authors Response:

Thanks for your suggestion. We prepared two additional Figures (3 and 4) for the revised manuscript.

Lines 169: Figure 3: Representative scheme describing how the nucleotides entrapped in liposomes are internalized by cells. As hydrophilic acids, the transport of DNA/RNA molecules is not so easy. The association of the nucleotides with liposomes may improve this internalization process.

Lines 277: Figure 4: Examples of different nucleotides and their biomedical applications described in this review.

We also included the following sequence to describe Figure 3 (lines 163-167): “For instance, nucleotide molecules are not well delivered to cells due to their high hydrophilicity. The encapsulation of these molecules in liposomes can improve this delivery, allowing them to reach the inner compartments of live cells. In Figure 3, we present a representative scheme with this process. While free molecules are not well absorbed, nucleotides entrapped in liposomes can pass throw the membrane more easily.”

  1. Examples of ultilization of these vesicles should be summarized as a table for the ease of reading.

Authors Response:

Thanks for your suggestion. To clarify this point, we included one individual table for each application presented in the revised manuscript. In total, we included 5 new tables, with the following information:

Reference

Nucleotide Carried

Target/Application 

Main Outcome

The new tables were inserted in the revised manuscript in the followings lines:

Line 370: Table 1: Liposomes for the delivery of nucleic acids in cancer therapy;

Line 424: Table 2: Liposomes for the delivery of nucleic acids for Immunostimulatory Effects;

Line 595: Table 3: Lipid-based ribozyme delivery systems for theranostic applications;

Line 632: Table 4: Veterinary application;

Line 705: Table 5: Nucleic acids and liposomes in the prevention of neglected tropical diseases.

  1. Typos and unfriendly mode of English usage can be found.

Authors Response:

Thanks for your review. We revised the full text with a native English reviewer and all orthographic and spelling improvements were included in the revised version of the manuscript. All revisions were highlighted in green.

Reviewer 3 Report

Given the vast literature on liposomes/lipid therapeutics, I think Authors should focus more on ODN, reproduce critical figures, present limitations and perspectives.

Authors need to describe the "other nucleotide transporters" and present their limits.

Authors need to discuss the immunogenicity of lipid carriers which was reported for liposomes and LNPs.

PAtisiran is an LNP not a liposome. Authors need to describe LNPs and their differences with liposomes in internal lipid organization.

Doxorubicin is not a nucleotide and out of the scope of the paper.

Authors need to reproduce critical figures form the cited papers and present the limits / disadvantages of the formulations in these papers.

Formulations of cited papers need to be presented.

Authors should cover antisense ODN and decoys.

Author Response

Reviewer 3

  1. Given the vast literature on liposomes/lipid therapeutics, I think Authors should focus more on ODN, reproduce critical figures, present limitations and perspectives.

Authors Response:

Thanks for your suggestion. As the reviewer mentioned, there is a vast literature on the field, and selecting which segment to go into deeply is not easy. Moreover, we understand that the use of nucleotides in different forms can revolutionize several biomedical fields. At the moment, is not easy to preview which one will have the best outcomes.  For now, we have a vast literature presenting the pre-clinical e initial proof-of-concept results, but the technology was applied more widely in a few real situations. Thus, for this review, we understand that a more narrow presentation is more suitable for the moment. To clarify this article's objective we included the following sentences in the introduction section (lines 266-275):

  1. 4. Review Objectives

As will be presented in this article review, the use of nucleotides and liposomes is vast and the biomedical applications are diverse. At the moment, is not easy to preview which biomedical applications will have the best benefits from using these technologies. Thus, we understand that, for the moment, we should not focus on a single application or nucleotide type, but rather present the different technological approaches available in the literature. For instance, we revised the main results regarding the use of nucleotide and liposome nanocarriers in different therapeutical and diagnostic applications, including oncology and immunological conditions, neglected diseases therapies, innovative diagnostics approaches, and veterinarian applications. These application examples are summarized in the following topics.

  1. Authors need to describe the "other nucleotide transporters" and present their limits.
  2. Authors need to discuss the immunogenicity of lipid carriers which was reported for liposomes and LNPs.

Authors Response:

Thanks for your suggestion. To clarify these two points, we included the following sentences in the revised manuscript (lines 79-86):

The development of these nanocarriers was also a response to the replacement of these other types of nucleotide carriers, which normally go through biotechnological manufacturing routes. For instance, the use of viral vectors may also contain highly immunogenic moieties that can induce adverse immunogenic reactions. This concern is also extended to lipid nanocarriers. At the moment the liposome immunogenicity seems to be related to the presence of molecules such as polyethylene glycol (PEG), which could be replaced in a near future by low immunogenic molecules. Additionally, it is possible to replace potential immunogenic lipid components or even remove them from the particle formulation.

  1. PAtisiran is an LNP not a liposome. Authors need to describe LNPs and their differences with liposomes in internal lipid organization.

Authors Response:

Thanks for your observation. To update and correct this paragraph, we included the following sentence in the revised version of our manuscript (Lines 143-146):

More recently, in 2018, the first lipid nanoparticle containing nucleotides were approved by the Food and Drug Administration (FDA) for amyloidosis treatment [18, 19]. Despite not being a true liposome, the development of lipid nanocarriers has increased the understanding of this type of nanostructure for the protection and delivery of nucleotides at target sites.

  1. Doxorubicin is not a nucleotide and out of the scope of the paper.

Authors Response:

Thanks for your observation. We included the doxorubicin example because the first liposome approved for pharmaceutical use was a product composed of Doxorubicin entrapped in liposome vesicles. Thus, we considered that this example could be used as a hallmark in liposome history. To clarify this argument, we included the following sentence in the revised manuscript (Lines 286-290):

As commented before, the first nanotech-based pharmaceutical product was a liposome containing doxorubicin, named Doxil™. For oncology, the use of liposomes is very common, and several examples use these lipid carriers to deliver different types of molecules, mainly chemotherapeutic drugs, such as doxorubicin, to target malignant tissues. Thus, the Doxil™ example is a hallmark in the history of liposome development.

  1. Authors need to reproduce critical figures form the cited papers and present the limits / disadvantages of the formulations in these papers.

Authors Response:

Thanks for your suggestion. We improved Figures 1 and 2 from the original paper; and included two additional figures (3 and 4) for the revised manuscript.

Lines 169: Figure 3: Representative scheme describing how the nucleotides entrapped in liposomes are internalized by cells. As hydrophilic acids, the transport of DNA/RNA molecules is not so easy. The association of the nucleotides with liposomes may improve this internalization process.

Lines 277: Figure 4: Examples of different nucleotides and their biomedical applications described in this review.

Reviewer 4 Report

The review written by Cardador et al on nucleotide entrapped liposome nanovesicles as a tool for therapeutic and diagnostic use is important.

It is well known that Compared with traditional drug delivery systems, liposomes exhibit better properties, including site-targeting, sustained or controlled release, protection of drugs from degradation and clearance, superior therapeutic effects, and lower toxic side effects.  Although this is not a new technology, as the authors mentioned it has become more important recently during COVID-19 vaccination.

I would author some points that can hopefully further improve this article.

1.      They can include details of liposomal products approved by the FDA and EMA.

2.      Although the authors have given details of the type of liposomes on page 1, it is brief. I suggest including a table that can categorize different types of available liposomes, their size, structure, and main composition.

3.      Authors have highlighted figures in the text, that need to be removed.

4.      Authors can illustrate the components used in the liposome-marketed products.

5.      No doubt, text is important, but to improve readability and enhance simplicity, more figures, flow charts, and illustrations will improve the citation.

6.      Remove the example from section 2, just write ‘ Theranostic Nucleotide.’

7.      Also, include some details about the manufacturing process used in the development of liposomes.

8.      Particle size is an important issue in liposomes, write something about particle size and size distribution.

9.      Including a section like future perspectives and recommendations will be good.

10.  Unfortunately, there is no section that has summarized all aspects discussed in the article, therefore, a section of discussion or summary could be introduced before giving a brief conclusion.

Author Response

Reviewer 4

  1. They can include details of liposomal products approved by the FDA and EMA.

Authors Response:

We included two examples of approved FDA lipid nanocarriers. One of them is a historical example, the Doxil (liposome containing Doxorubicin); the second was a lipid nanocarrier named Partisiran; and we also cited the mRNA COVID-19 vaccines as an example of lipid nanoparticles approved by FDA. To clarify these examples, we updated the sentences related to these points:

Lines 272-276: As commented before, the first nanotech-based pharmaceutical product was a liposome containing doxorubicin, named Doxil™. For oncology, the use of liposomes is very common, and several examples use these lipid carriers to deliver different types of molecules, manly chemotherapeutic drugs, such as doxorubicin, to target malignant tissues. Thus, the Doxil™ example is a hallmark in the history of liposome development.

Lines 154-156: These liposome-mRNA vaccines, which were approved by different regulatory agencies worldwide, have been by far the nanomedical product most widely used by humans.

(Lines 140-143): More recently, in 2018, the first lipid nanoparticle containing nucleotides were approved by the Food and Drug Administration (FDA) for amyloidosis treatment [18, 19]. Despite not being a true liposome, the development of lipid nanocarriers has increased the understanding of this type of nanostructure for the protection and delivery of nucleotides at target sites.

  1. Although the authors have given details of the type of liposomes on page 1, it is brief. I suggest including a table that can categorize different types of available liposomes, their size, structure, and main composition.

Authors REsposnse:

Thanks for your suggestion. To clarify this point, we included the 5 new tables in the revised manuscript. The new tables were inserted in the revised manuscript in the followings lines:

Line 370: Table 1: Liposomes for the delivery of nucleic acids in cancer therapy;

Line 424: Table 2: Liposomes for the delivery of nucleic acids for Immunostimulatory Effects;

Line 595: Table 3: Lipid-based ribozyme delivery systems for theranostic applications;

Line 632: Table 4: Veterinary application;

Line 705: Table 5: Nucleic acids and liposomes in the prevention of neglected tropical diseases.

  1. Authors have highlighted figures in the text, that need to be removed.

Authors response:

Thanks, we removed the yellow highlight was removed in the revised manuscript

  1. Authors can illustrate the components used in the liposome-marketed products.
  2. No doubt, text is important, but to improve readability and enhance simplicity, more figures, flow charts, and illustrations will improve the citation.

Authors response:

To clarify these two points we included two new Figures (3 and 4) and revised the original Figures 1 and 2. The revised and new figures were placed in the following lines in the revised manuscript:

Lines 169: Figure 3: Representative scheme describing how the nucleotides entrapped in liposomes are internalized by cells. As hydrophilic acids, the transport of DNA/RNA molecules is not so easy. The association of the nucleotides with liposomes may improve this internalization process.

Lines 277: Figure 4: Examples of different nucleotides and their biomedical applications described in this review.

  1. Remove the example from section 2, just write ‘ Theranostic Nucleotide.’

Authors' response: We removed the word example.  

  1. Also, include some details about the manufacturing process used in the development of liposomes.

Authors response:

Dear reviewer, this is a very complex issue, especially for commercial examples of liposomes. We considered that this kind of discussion could be placed in a new article, dedicated specifically to this topic. This is an important discussion in the literature, and a full review of the theme may attract the attention of the audience. Thus, we considered that our review is not the specific space for such discussion.

  1. Particle size is an important issue in liposomes, write something about particle size and size distribution.

Authors response:

Dear reviewer, we considered using this information, however, there are different administration routes used for each specific application. For instance, intramuscular vaccines usually have wider liposome vesicles, and PDI is not a critical issue. On the other hand, intravenous administration is much more critical for the liposome vesicles. Thus, we considered that for this review it should be better to not present this information that could confuse the reader.

  1. Including a section like future perspectives and recommendations will be good.

Authors response:

Thanks for your suggestion. We include a perspective section with the personal author's view of the field. The idea was to share how we observe the evolution of these technologies for the applications that were discussed during the review. We included the following sentence in lines 717-726:

Perspectives for Biomedical Applications

In addition to these technical perspectives, the development of these technologies using nucleotides as therapeutic and/or diagnostic molecules has a great chance of completely revolutionizing the treatment and even the control of several diseases of biomedical interest, as presented during the present review. In terms of historical perspectives, the incorporation of these technologies can be ranked as one of the greatest advances in the biomedical area of the century. Perhaps, we can compare it to the development of antibiotics, which is considered one of the greatest advances in medicine in the last century. For sure, the COVID-19 pandemic was the big challenge faced by these technologies, however, other super important issues, such as cancer, neglected diseases, and food production may also be benefited from this technological revolution.

  1. Unfortunately, there is no section that has summarized all aspects discussed in the article, therefore, a section of discussion or summary could be introduced before giving a brief conclusion.

Authors Response:

Thanks for your suggestion. We included 5 new tables to summarize the main examples used during each sub-section. The new tables were included in:

Line 370: Table 1: Liposomes for the delivery of nucleic acids in cancer therapy;

Line 424: Table 2: Liposomes for the delivery of nucleic acids for Immunostimulatory Effects;

Line 595: Table 3: Lipid-based ribozyme delivery systems for theranostic applications;

Line 632: Table 4: Veterinary application;

Line 705: Table 5: Nucleic acids and liposomes in the prevention of neglected tropical diseases.

Round 2

Reviewer 2 Report

The manuscript has been improved to a certain extent, except the followings.

- The information in the Tables are too simple. For instance, the authors should give more details on the cargo loaded, e.g. siRNA targeting which gene, instead of just simply mention the sRNA only.

- Figure 1, it is unclear what is "Individual structure" ? The cartoon just looks like a single phospholipid and it is unclear what special of this to be emphasized.

- Typos and unfriendly mode of English usage can still be found. For example, in the last paragraph, 

"We understand, based on the literature references that several solutions, based on nucleotide entrapped in cationic liposomes will be developed and made available to society."

there are too frequent use of "based on" in one sentence. The authors should proofread the whole manuscript again to avoid these redundant wordings.

Author Response

Minor Revisions – March 2th 2023

Dear Editors,

First, I would like to thank you for the opportunity to revise our submitted review article to Pharmaceutics. We analysed and revised all the suggested revisions provided by the manuscript reviewer. We prepared individual answers for each point and included the revisions in the main manuscript. To clarify these modifications, we highlighted in yellow all the text inclusions. Additionally, some orthographic modifications were also revised in the article. Regarding these orthographic modifications, we kept the track changes highlighted.

On behalf of the Article authors, João Paulo Longo

Reviewer Report (round 2):

The manuscript has been improved to a certain extent, except the followings.

  1. The information in the Tables are too simple. For instance, the authors should give more details on the cargo loaded, e.g. siRNA targeting which gene, instead of just simply mention the sRNA only.

Authors Response:

Thanks for your suggestion. To clarify this point, we included the molecule or gene that the liposome entrapped nucleotide is targeting. All the inclusions are highlighted in yellow.

Table 1: Liposomes for the delivery of nucleic acids in cancer therapy

Reference

Nucleotide Carried

Target Disease

Main Outcome

Aleku et al. 2008

siRNA (Atu027), targets the expression of protein kinase N3 (PKN3)

Pre-clinical -Prostate and Pancreatic Cancer

Reduction in tumor vessels density

Schultheis et al. 2016

siRNA (Atu027), targets the expression of protein kinase N3 (PKN3)

Clinical Trial - Advanced pancreatic carcinoma

Atu027, in combination with gemcitabine, was well tolerated in the clinical trial. The results suggest the efficacy of Atu027 in the pancreatic carcinoma therapy

Zhuo et al. 2018

pcDNA3.1-CSF1-endostatin, mouse endostatin gene

Pre-clinical Mouse/breast cancer

Tumor suppression by 71%

Mu et al. (2020)

Herpes simplex virus thymidine kinase/ganciclovir (HSVtk/GCV) suicide gene system

Pre-clinical - mice hepatocellular carcinoma

Suppressed tumor growth and extended mice survival time

Zhao et al. (2015)

siRNA, for silencing c-Myc and VEGF oncogenic pathways

Pre-clinical - mice lung cancer

siRNA silenced distinct oncogenic pathways (c-MYC and VEGF) in mice lung tumors

Zhu et al. (2019)

siRNA, IGF-1R-siRNA to inhibit tumor growth

Pre-clinical - mice lung cancer

Controlled lung tumor growth tumor-bearing mice

Table 2: Liposomes for the delivery of nucleic acids for Immunostimulatory Effects

Reference

Nucleotide Carried

Target/Application 

Main Outcome

Kranz et al. (2016)

RNA, RNA-Lipoplex that triggers interferon-α (IFNα) release

Macrophages and Dendritic Cells

Effector and memory T-cell responses were triggered, and interferon-α-dependent rejection of tumors was observed in vivo

Lorentzen et al. (2022)

mRNA – review article with different vaccine applications

Vaccine against melanoma

Induced durable objective responses in patients with unresectable melanoma.

Gursel et. al (2001)

CpG oligonucleotides, unmethylated oligodeoxynucleotides containing CpG motifs like bacterial nucleotides that trigger immune cells

Immune adjuvants, anti-allergens, and immunoprotective agents

Increased 15- to 40-fold antigen immunization in mice models

Loira-Pastoriza et. al (2021)

CpG oligonucleotides, unmethylated oligodeoxynucleotides containing CpG motifs similar to bacterial nucleotides that trigger immune cells

Immunoadjuvant for murine lung tumors

Liposomes increased CpG effectiveness in controlling murine lung tumors

Guan et al (2018)

Spherical nucleic acids, RNA selective for toll-like receptors (TLRs) 7/8 targgeting

Dendritic Cells

Potently activate TLR7/8 via NF-κΒ signaling

Callmann (2020)

Spherical nucleic acids, are oligonucleotides functionalized with hydrophobic carbon chains to increase their liposome encapsulation. In this report, authors used immunostimulatory  oligonucleotides

Pre-clinical – Murine Breast  Cancer

Reduce both primary tumor and metastatic growth due to immunostimulant effects

Table 3: Lipid-based ribozyme delivery systems for theranostic applications

Reference

Nucleotide Carried

Target/Application 

Main Outcome

Sioud et al., 1992

Hammerhead ribozyme-mediated suppression of TNF-α through transfection with cationic liposomes.

HL60 and PBMNC cell lines in models of inflammatory rheumatic diseases.

90% and 85% reduction of tumor necrosis factor alpha mRNA and protein, respectively.

Sioud, 1996

Suppression of LPS-induced TNF-α through hammerhead ribozyme-mediated transfection with cationic liposomes.

Mice/ Inflammatory rheumatic disease

50-70% inhibition of TNF-α gene expression through ribozymes.

Snyder et al., 1993

Hammerhead ribozymes with RNA and RNA-DNA hybrid structures. Ribozyme structure cleaved bcr-abl (abnormal fused gene present in chronic leukemia) mRNA

EM-2 cell line and patient-derived cell lines with blast crisis CML.

Reduced levels of bcr-abl mRNA involved in the pathogenesis of Ph1+ Leukemia.

Leopold et al., 1995

Transfection of double- and triple-unit ribozymes using liposomes or folic acid-polylysine as carriers.

32D cells from murine myeloblasts associated with CML model disease.

Reduced levels of bcr-abl mRNA from the tyrosine kinase fusion gene resulting from the bcr gene on chromosome 22 and abl gene on chromosome 9.

Jarvis et al., 1996

Hammerhead ribozymes complexed with DOSPA. Cleave c-myb RNA

Rat aortic smooth muscle cells from female rats.

Reduction of c-myb proto-oncogene mRNA associated with the proliferation of vascular smooth muscle cells.

Scherr et al., 1997

Hammerhead ribozymes complexed with DOSPA. Targeting  N-ras oncogene

HeLa cells assays

Cleavage of N-ras oncogene RNAm in HeLa cells and reduced expression of N-ras/luciferase fusion gene.

Sioud and Sørensen, 1998

The complex of modified PKCα ribozyme with cationic liposome. Inhibition of protein kinase Cα

Solid tumors of malignant glioma.

Inhibition of PKCα gene expression.

Kossen et al., 2004

Allosteric Half-ribozyme used to detect natural viral sequences variants

Detect the hepatitis C virus.

Upon binding to the target RNA the half-ribozyme would go through a conformational change that allows substrate one with biotin on its 3’ and substrate two carrying a 5’-fluorescein to bind together. being directly detected or its signal could be amplified

Penchovsky, 2012

Allosteric hammerhead ribozyme

Detect oculopharyngeal muscular dystrophy (OPMD) disorder

Allosteric hammerhead ribozymes, which undergo a conformational change to their active forms in the presence of the RNA-carrying (GCG)11 repeats transcribed from the gene that encodes the human polyadenylate-binding nuclear protein 1 (PABPN1)

Liao et al., 2018

Aptamer, and an allosteric hammerhead ribozyme

Detect the bronchodilator theophylline

Upon binding of the ligand-target and the aptamer, the ribozyme under a conformational change and self-cleavage triggers the amplification of a reporter, which oxidizes a substrate leading to a visible color change

Table 4: Veterinary applications

Reference

Nucleotide Carried

Target/Application 

Main Outcome

Camussone et al (2022)

α-toxin gene  (plasmid).

CpG oligonucleotides (unmethylated oligodeoxynucleotides containing CpG motifs like bacterial nucleotides that trigger immune cells)

Immunization Against S. aureus intramammary infection (cow mastitis)

Reduction, but not significant of animal mastitis

Liu et al (2022)

DNA plasmid (infectious anemia virus sequences)

The plasmid was used as a vaccine booster in combination with the antigen exposition

Chicken viral infectious anemia

DNA vaccine alone did not protect against the infection. The combined vaccine (plasmid + antigen) had the best protection against the infection.

Mueller et al (2005)

DNA plasmid containing

CpG oligonucleotides, which are unmethylated oligodeoxynucleotides that trigger immune cells

Immunostimulant against refractory canine atopic dermatitis

The vaccine decreased some signs and symptoms related to atopic dermatitis, as well as a reduction in IL-4 production

Ommen et al (2022)

DNA aptamers that targets and bound to S. aureus cells

The DNA aptamers were used as targeting molecules for liposome delivery to S. aureus biofilms

The aptamer was useful to improve the delivery of antibiotics entrapped in nano-sized liposomes

Lee et al (2017)

Proprietary oligonucelotides (F5881, F5506, T2684, Sigma-Aldrich, St Louis, MO)

Adjuvants in polyclonal antibody production in rabbits

The liposome oligonucleotide complexes were effective as immune adjuvants for polyclonal antibody production in rabbits

Table 5: Nucleic acids and liposomes in the prevention of neglected tropical diseases

Reference

Nucleotide Carried

Target/Application 

Main Outcome

Yin et al., 2021

Hepatitis B vírus plasmids expressing HBcAg and HBeAg

Hepatitis B  (vaccine)

Strong Th1 and Th2 immune response resulting in the elimination of the virus after the challenge

Rodríguez-Morales et al., 2012

Plasmids containing Trypanosoma cruzi genes encoding the Tc SP (trans-sialidase protein) and Tc SSP4 expression(amastigote-specific protein)

Chagas disease (vaccine)

Induction of moderate level of protection in immunized dogs

Li et al., 2011

Plasmid containing the gene encoding glutathione S-transferase of Schistosoma japonicum

Schistosomiasis (vaccine)

The reduction rate of worm and egg burdens in the pEGFP-Sj26GST plus CIM group was more than  68 % and  higher than that in pEGFP-Sj26GST alone ( P<0.01)

Wollner et al., 2021

Nucleotide-modified mRNA vaccine encoding the membrane and envelope structural proteins from DENV serotype 1 encapsulated in lipid nanoparticles

Dengue (vaccine)

Robust antiviral immune responses comparable to viral infection, with high levels of neutralizing antibody titers and antiviral CD4+ and CD8+ T cells

Bollman et al., 2022

mRNA of VLPs Zika virus. mRNA-1325 encodes Zika Virus membrane envelope proteins

Zika (vaccine)

Complete protection against ZIKV challenge in non-human primates

  1. Figure 1, it is unclear what is "Individual structure" ? The cartoon just looks like a single phospholipid and it is unclear what special of this to be emphasized.

Authors Answer:

Thanks for your suggestion. The "Individual structure" mentioned in the figure legend referred to the individual structure of a phospholipid that constitutes the liposome of interest. It was named an "individual structure" because it was not a specific phospholipid structure. However, for better understanding, Figure 1 has been altered and the SUV structure is now constituted by cationic phospholipids, in pink, as explained in the legend.

The purpose of figure 1 is to emphasize the diversity of size, layers, and agents of interest that can be entrapped in the liposomes. For example, now Figure 1 shows hydrophilic molecules entrapped in the aqueous core of the LUV, while the hydrophobic ones are protected in the bilayer presented. Moreover, the representation of the liposomes with both regular phospholipids and cationic phospholipids in the same vesicles:

  1. Typos and unfriendly mode of English usage can still be found. For example, in the last paragraph: "We understand, based on the literature references that several solutions, based on nucleotide entrapped in cationic liposomes will be developed and made available to society."

Authors answer:

Thanks for your review. To improve this last sentence, we replace the phrase by the following (Lines 716-717): “Thus, a variety of solutions that rely on nucleotides entrapped in cationic liposomes will be created and offered to the public in the next years.”

  1. there are too frequent use of "based on" in one sentence. The authors should proofread the whole manuscript again to avoid these redundant wordings.

Authors answer:

Thanks for your suggestion. We revised the revised manuscript and modified the paragraphs to replace the “base on” sentences. All the replacements are highlighted in yellow in the revised manuscript.
